# DOMAIN-WISE ADVERSARIAL TRAINING FOR OUT-OF-DISTRIBUTION GENERALIZATION

## ABSTRACT

Despite the impressive success on many tasks, deep learning models are shown to rely on spurious features, which will catastrophically fail when generalized to out-of-distribution (OOD) data. To alleviate this issue, Invariant Risk Minimization (IRM) is proposed to extract domain-invariant features for OOD generalization. Nevertheless, recent work shows that IRM is only effective for a certain type of distribution shift (*e.g.,* correlation shift) while fails for other cases (*e.g.,* diversity shift). Meanwhile, another thread of method, Adversarial Training (AT), has shown better domain transfer performance, suggesting that it is potential to be an effective candidate for extracting domain-invariant features. In this paper, we investigate this possibility by exploring the similarity between the IRM and AT objectives. Inspired by this connection, we propose Domain-wise Adversarial Training (DAT), an AT-inspired method for alleviating distribution shift by domain-specific perturbations. Extensive experiments show that our proposed DAT can effectively remove the domain-varying features and improve OOD generalization under both correlation shift and diversity shift.

## 1 INTRODUCTION

Modern deep learning techniques have achieved remarkable success on many tasks (He et al., 2016; Brown et al., 2020). Yet, under some scenarios, deep models will suffer a catastrophic performance degradation since they tend to seize on spurious correlations in the training data (Beery et al., 2018). One of those representative scenarios is the Out-of-Distribution (OOD) generalization, where one expects the trained model to perform well at the test time even when the training and testing data come from different distributions[1]. Another representative scenario under which deep models are unstable is the adversarial example. Researchers have found that deep models are quite brittle since one can inject imperceptible perturbations into the input and cause the model to make wrong predictions with extremely high confidence (Szegedy et al., 2014).

These two issues have some similarities to each other. They both arise because deep networks do not learn the essential causal associations (or intrinsic features). Nevertheless, in their corresponding fields, different approaches have been proposed. For OOD generalization, a large class of methods called Invariant Causal Prediction (ICP) (Peters et al., 2016) are proposed. Among them, Invariant Risk Minimization (IRM) (Arjovsky et al., 2019) attracts significant attention, which intends to extract features that are invariant across different data distributions and expects the model to ignore information related to the environment. While for adversarial robustness against adversarial examples, Adversarial Training (AT) (Madry et al., 2018) is the most effective approach at the current stage (Athalye et al., 2018). It trains a model on adversarial examples that are generated by injecting perturbations optimized for each image into natural examples. These two fields seem to be independent, and their connections are rarely studied. That is exactly what we are exploring in this paper.

Although IRM and its variants have shown promise on certain tasks, *e.g.,* CMNIST (Arjovsky et al., 2019), recent studies (Gulrajani & Lopez-Paz, 2021) show that on a large-scale controlled experiment on OOD generalization, all these methods fail to exceed the simplest *i.i.d.* baseline, *i.e.,* Empirical Risk Minimization (ERM). Through a dissection of the benchmark datasets, Ye et al. (2021)

---

[1] In current literature, the terminologies of "domain", "environment" and "distribution" are often used interchangeably, so we do the same in the whole paper.

notice that there are actually two types of distribution shifts: correlation shift (same support, different correlation) and diversity shift (different support, same correlation). IRM variants can only perform well under (some) correlation shift while performing poorly under diversity shift. Thus, we need to seek better alternatives for OOD generalization, while AT seems to be a promising candidate from both theoretical and empirical aspects. Theoretically, by learning invariance *w.r.t.* local input perturbations, AT can be regarded as Distributionally Robust Optimization (DRO) (Sinha et al., 2018; Volpi et al., 2018; Rahimian & Mehrotra, 2019; Duchi et al., 2021) over $\ell_p$-bounded distributional shift. Thus, AT could reliably extract robust features, *e.g.,* the object shape, from the input. Empirically, several recent works show that AT has better domain transferability than ERM (Salman et al., 2020). These findings naturally leads to the questions:

*Will AT perform better than IRM? Will AT be helpful for OOD generalization?*

In this paper, we take a further step to answering these intriguing questions. We first reveal the connections between IRM and AT, and find that IRM can be regarded as an instance-reweighted version of Domain-wise Adversarial Training (DAT), a new version of adversarial training that we propose for generalizing Universal Adversarial Training (Moosavi-Dezfooli et al., 2017) to multiple domains. Inspired by this connection, we further explore how DAT performs for OOD data. We first notice that DAT is suitable for solving domain generalization problems as it can effectively remove the relatively static background information with domain-wise perturbations. We further verify this intuition on both synthetic tasks (Xiao et al., 2021) and real-world datasets, where we show clear advantages over ERM. At last, we conduct extensive experiments on benchmark datasets and show our DAT can outperform ERM consistently on tasks dominated by both correlation shift and diversity shift, and as a result, achieve state-of-the-art performance (in average) on these datasets.

We summarize our contributions as follows:

- We develop a new kind of adversarial training, Domain-wise Adversarial Training (DAT), for domain generalization, and we establish the intrinsic similarity between IRM and domain-wise AT objectives.
- We analyze how DAT will benefit learning invariant features and verify our hypothesis through both synthetic data and real-world datasets.
- Extensive experiments on benchmark datasets show that DAT does not only perform better than ERM under correlation shift like IRM but also outperforms ERM under diversity shift like (sample-wise) AT. Therefore, our methods achieve state-of-the-art results by surpassing ERM at both kinds of distribution shifts.

## 2 RELATED WORKS

**IRM and Its Variants**  Invariant Risk Minimization (IRM) (Arjovsky et al. (2019)) develops a paradigm to extract causal (invariant) features and find the optimal invariant classifier on top over several given training environments. The work of Kamath et al. (2021) reveals the gap between IRM and IRMv1, show that even in a simple model that echos the idea of the original IRM objective, IRMv1 can fail catastrophically. Rosenfeld et al. (2021) prove that when the number of training environments is not large enough, IRM can face the risk of using environmental features. There also exists a predictor feasible for IRMv1 that is very similar to the ERM solution.

**AT and Its Variants**  Szegedy et al. (2014) report one can inject imperceptible perturbations to fool deep models. Athalye et al. (2018) reveal that among the proposed defenses, adversarial training was the only effective one. Adversarial Training (AT) (Goodfellow et al., 2014; Madry et al., 2018) is the representative approach to train robust models. Recently, Kamath et al. (2021) show adversarially learned features can transfer better than standardly trained models, while various works (Volpi et al., 2018; Sinha et al., 2018; Ford et al., 2019; Qiao et al., 2020; Yi et al., 2021; Gokhale et al., 2021) adopt sample-wise adversarial training or adversarial data augmentation to improve OOD robustness. However, most of the discussions are limited to distributional robustness *w.r.t.* Wasserstein distance, making it less practical for accounting real-world OOD scenarios as discussed in this work.

**Evaluating OOD Robustness**  Gulrajani & Lopez-Paz (2021) recently noticed that the evaluation criteria are crucial for fair comparison of OOD robustness, where under fair settings, no

algorithm has performed consistently better than vanilla ERM. Ye et al. (2021) further identifies two kinds of distribution shifts in current benchmark datasets: diversity shift and correlation shift. Diversity shift refers to the shift of the distribution support of spurious feature $z$, while correlation shift refers to the change of conditional probability of label $y$ given spurious feature $z$ on the same support. We present an illustrative example in Figure 1. In particular, they show that there seems to be a trade-off between the two tasks as an algorithm that performs well at one task tends to perform poorly on the other.

Instead, in this work, we show that our DAT can achieve superior performance on both correlation shift and diversity shift tasks.

**Universal Adversarial Training** Universal adversarial perturbation proposed by Moosavi-Dezfooli et al. (2017) is a type of adversarial attack that adopts a universal perturbation for all images. Universal adversarial training (UAT) (Shafahi et al., 2020) is then proposed to defend against this attack by training against universally perturbed data. Instead, in our work, we firstly show that we can adapt UAT for solving domain generalization problems.

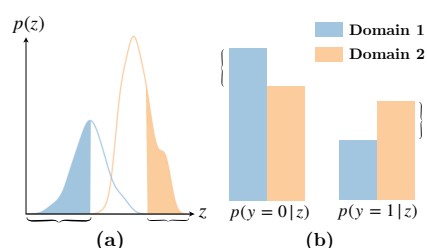

(a)        (b)

Figure 1: An illustrative example of the two kinds of distribution shifts. (a) Diversity shift (b) Correlation shift. $z$ stands for spurious feature, and $y$ stands for label class.

## 3 ON THE RELATIONSHIP BETWEEN IRM AND AT

### 3.1 PRELIMINARY

**Notation** Let $\Phi : \mathcal{X} \subset R^n \to R^d$ denotes the representation of a $\theta$-parameterized piecewise linear classifier, *i.e.*, $\Phi(\cdot) = \phi^L(W^L\phi^{L-1}(\dots) + b^{L-1}) + b^L$, where $\phi^L$ is the activation function, and $W^L, b^L$ denote the layer-wise weight matrix and bias vector, collectively denoted by $\theta$. Additionally, let $\beta$ be the linear classifier on top, and the output of the network is $\beta \cdot \Phi(x) = \beta^\top \Phi(x)$. Let $\ell(\hat{y}, y) = -\log \sigma(y\hat{y})$ be the sample logistic loss. We consider a two-class ($y = \pm 1$) classification setting with output dimension $d = 1$, and our discussion can be easily extended to the general cases.

**ERM** The traditional Empirical Risk Minimization (ERM) algorithm optimizes over the loss on i.i.d. data, *i.e.*,

$$\min_{\beta, \Phi} R(\beta \cdot \Phi), \text{ where } R(\beta \cdot \Phi) = \mathbb{E}_{(x,y)\sim D}\ell(\beta^\top \Phi(x), y). \tag{1}$$

In OOD generalization problem, one faces a set of (training) *environments* $\mathcal{E}$, where each environment $e \in \mathcal{E}$ corresponds to a unique data distribution $D_e$. When facing multiple environments, the ERM objective simply mixes the data together and takes the form

$$(\text{ERM}) \quad \min_{\beta, \Phi} \sum_e R^e(\beta \cdot \Phi), \text{ where } R^e(\beta \cdot \Phi) = \mathbb{E}_{(x,y)\sim D_e}\ell(\beta^\top \Phi(x), y). \tag{2}$$

**IRM and IRMv1** Instead of simply mixing the data together, IRM seeks to learn an *invariant* representation $\Phi$ such that it can be minimizing with the same classifier $\beta$. Formally we have

$$(\text{IRM}) \quad \begin{aligned} &\min_{\beta, \Phi} \sum_{e\in\mathcal{E}} R^e(\beta \cdot \Phi) \\ &\text{s.t. } \beta \in \arg\min_{\bar{\beta}} R^e(\bar{\beta} \cdot \Phi), \forall e \in \mathcal{E}. \end{aligned} \tag{3}$$

Since this bi-level optimization problem is difficult to solve, the practical version IRMv1 as regularized ERM, where the gradient penalty is calculated *w.r.t.* a dummy variable $w$:

$$(\text{IRMv1}) \quad \min_{\beta, \Phi} \sum_{e\in\mathcal{E}} \Big[ R^e(\beta \cdot \Phi) + \lambda \cdot \underbrace{||\nabla_{w|w=1.0} R^e(w \cdot (\beta \cdot \Phi))||^2}_{\text{Penalty}_{\text{IRM}}} \Big]. \tag{4}$$

**AT** Adversarial Training (Madry et al., 2018)) instead replaces ERM with a minimax objective,

$$\text{(AT)} \quad \min_{\beta, \Phi} R^{\text{AT}}(\beta \cdot \Phi) = \min_{\beta, \Phi} \mathbb{E}_{(x,y) \sim D} \max_{||\delta_x||_p \leq \varepsilon} \ell(\beta^\top \Phi(x + \delta_x), y), \tag{5}$$

where one maximizes the inner loss by injecting *sample-wise* perturbations $\delta_x$ and solve the outer minimization *w.r.t.* parameters $\beta, \Phi$ on the perturbed sample $(x + \delta_x, y)$. Typically, the perturbation is constrained within an $\ell_p$ ball with radius $\varepsilon$. In this way, AT can learn models that are robust to $\ell_p$ perturbations.

**UAT** Instead of injecting sample-wise perturbations, Moosavi-Dezfooli et al. (2017) notice that we can also adopt a *universal* perturbation $\delta$ for all samples, which results in the Unversarial Adversarial Training (UAT) objective as follows:

$$\text{(UAT)} \quad \min_{\beta, \Phi} R^{\text{UAT}}(\beta \cdot \Phi) = \min_{\beta, \Phi} \max_{||\delta||_p \leq \varepsilon} \mathbb{E}_{(x,y) \sim D_e} \ell(\beta^\top \Phi(x + \delta), y), \tag{6}$$

### 3.2 RELATING ADVERSARIAL TRAINING TO IRM

**Motivation** As shown above, it seems that IRM and AT are two distinct learning paradigms, while in fact, we can show that IRM is closely related to a certain kind of adversarial training. To see this, we first notice that AT can be rephrased into a regularized ERM loss with a penalty on *sample-wise* robustness through linearization:

$$
\begin{aligned}
R^{\text{AT}}(\beta \cdot \Phi) &= \mathbb{E}_{(x,y) \sim D} \max_{||\delta_x|| \leq \varepsilon} \ell(\beta^\top \Phi(x + \delta_x), y) \\
&= \mathbb{E}_{(x,y) \sim D} \Big[ \ell(\beta^\top \Phi(x), y) + \max_{||\delta_x|| \leq \varepsilon} \big( \ell(\beta^\top \Phi(x + \delta_x), y) - \ell(\beta^\top \Phi(x), y) \big) \Big] \\
&\approx R(\beta \cdot \Phi) + \varepsilon \underbrace{\mathbb{E}_{(x,y) \sim D} \big\| \nabla_x \ell(\beta^\top \Phi(x), y) \big\|}_{\text{Penalty}_{\text{AT}}},
\end{aligned}
\tag{7}
$$

which resembles the gradient penalty adopted in IRMv1. One main difference is that AT's penalty is calculated *w.r.t.* sample-wise gradients, while IRM's penalty *w.r.t.* the population loss. This difference motivates us to adopt a population-level perturbation $\delta$ instead, which, in the literature of adversarial learning, corresponds to Universal AT (Eq. 6) that could be rephrased as follows:

$$
\begin{aligned}
R^{\text{UAT}}(\beta \cdot \Phi) &= \max_{||\delta|| \leq \varepsilon} \mathbb{E}_{(x,y) \sim D} \ell(\beta^\top \Phi(x + \delta), y) \\
&= \mathbb{E}_{(x,y) \sim D} \ell(\beta^\top \Phi(x), y) + \max_{||\delta|| \leq \varepsilon} \mathbb{E}_{(x,y) \sim D} \big( \ell(\beta^\top \Phi(x + \delta), y) - \ell(\beta^\top \Phi(x), y) \big) \\
&\approx R(\beta \cdot \Phi) + \varepsilon \underbrace{\big\| \nabla_x \mathbb{E}_{(x,y) \sim D} \ell(\beta^\top \Phi(x), y) \big\|}_{\text{Penalty}_{\text{UAT}}}.
\end{aligned}
\tag{8}
$$

**DAT** Inspired by the connection above between UAT with IRM, we adapt UAT to the OOD setting with multiple domains and propose Domain-wise Adversarial Training (DAT), which adopts a *domain-wise* perturbation $\delta_e$ for each training domain $e \in \mathcal{E}$. Formally, we have

$$
\begin{aligned}
&\min_{\beta, \Phi} \sum_{e \in \mathcal{E}} \mathbb{E}_{(x,y) \sim D_e} \ell \big( \beta^\top \Phi(x + \delta_e)), y \big) \\
&\text{s.t. } \delta_e \in \arg\max_{||\delta|| \leq \varepsilon} \mathbb{E}_{(x,y) \sim D_e} \ell(\beta^\top \Phi(x + \delta), y), \forall e \in \mathcal{E}.
\end{aligned}
\tag{9}
$$

The objective could be interpreted as the loss function of universal training in a domain-wise fashion, which lead to solving for the perturbation vector individually for each domain. We name this method as Domain-wise Adversarial Training (DAT), a generalization of UAT for learning invariance from multiple domains. In practice, we solve the problem above with alternating update of model parameters $\beta, \Phi$ and perturbations $\delta_e$. Specifically, for each mini-batch $B_e$ sampled from domain $\mathcal{D}_e$, we update $\delta_e$ with $B_e$ using gradient ascent to find the best adversarial perturbations. Then the adversarial samples are used to train the model. A detailed description is shown in Algorithm 1.

---

**Algorithm 1** Domain-wise Adversarial Training

---

**Input:** Dataset of multiple environments $D_e, e \in \mathcal{E}$, desired $l_p$ norm of the perturbation $\varepsilon$, perturbation step size $\alpha$
**Output:** Model $(\Phi, \beta)$
  Randomly initiate $\theta$, perturbation $\delta_e, \forall e \in \mathcal{E}$
  **for** iterations $= 1, 2, 3, \dots$ **do**
    **for** each environment $e$ **do**
        1. Sample batch $B_e$ from environment $e$
        2. Update the perturbation $\delta_e$ using one-step gradient ascent with step size $\alpha$
        3. Project the perturbation $\delta_e$ to the $\ell_p$ ball of radius $\varepsilon$
        4. Generate adversarial examples $x_{adv} \leftarrow x + \delta_e, \forall x \in B_i$
        5. Update $\Phi$ and $\beta$ with gradient descent (Adam) on $x_{adv}$
    **end for**
  **end for**

---

### 3.3 Connection between IRM and DAT

Here, we establish a formal connection between IRM and DAT. To begin with, we note that DAT can also be reformulated as a regularized ERM in the multi-domain scenario.

$$R^{\mathrm{DAT}}(\beta \cdot \Phi) = \sum_{e \in \mathcal{E}} \max_{||\delta_e|| \leq \varepsilon} \mathbb{E}_{(x,y) \sim D_e} \ell(\beta^\top \Phi(x + \delta_e), y)$$

$$= \sum_{e \in \mathcal{E}} \left[ \mathbb{E}_{(x,y) \sim D_e} \ell(\beta^\top \Phi(x), y) + \max_{||\delta_e|| \leq \varepsilon} \mathbb{E}_{(x,y) \sim D_e} \left( \ell(\beta^\top \Phi(x + \delta_e), y) - \ell(\beta^\top \Phi(x), y) \right) \right]$$

$$\approx \sum_{e \in \mathcal{E}} \left[ R^e(\beta \cdot \Phi) + \varepsilon \underbrace{\left\| \nabla_x \mathbb{E}_{(x,y) \sim D_e} \ell(\beta^\top \Phi(x), y) \right\|}_{\mathrm{Penalty_{DAT}}} \right].$$

$$(10)$$

Based on this reformulation, we can show that there exists an intrinsic relationship between DAT and IRM as in the following proposition:

**Proposition 1** *Consider each $D_e$ as the corresponding distribution of a particular training domain $e$. For any $\beta \cdot \Phi$ as a deep network with any activation function, the penalty term of IRMv1,* $\mathrm{Penalty_{IRM}}$ *(Eq. 4), could be expressed as the square of a reweighted version of the penalty term of the above approximate target,* $\mathrm{Penalty_{DAT}}$*(Eq. 10), on each environment $e$ with coefficients related to the distribution $D_e$, which could be stated as follows:*

$$\mathrm{Penalty_{IRM}} = \left\| \mathbb{E}_{D_e}[L_x x] \right\|^2, \text{ and } \mathrm{Penalty_{DAT}} = \left\| \mathbb{E}_{D_e} L_x \right\| \qquad (11)$$

*where $L_x = (1 - \sigma(y\beta^\top \Phi_x x)) y\beta^\top \Phi_x$.*

The proposition above shows that IRM can be regarded as an instance-reweighted version of DAT, which indicates that DAT can also be applied to improve out-of-distribution generalization performance. We explore this possibility in the next section.

Besides, assume that each domain only contains one sample, we can further obtain a stronger result, that is, the equivalence between IRM and (linearized) DAT as follows.

**Remark 1 (Equivalence under Single-sample Environments)** *When the environments degenerate into a single data point, we have the following relationship: If $\varepsilon$ is sufficiently small, then for $\beta \cdot \Phi$ as a deep network with any activation function, the penalty term of IRMv1 (Eq. 4) on each sample and the square of the maximization term of Linearized version of Eq. 9 (LDAT, obtained by first-order approximation of DAT)*

$$\mathrm{Penalty_{LDAT}} = \left\langle \nabla_x \ell \left( \beta^T \Phi(x), y \right), \pm \hat{\delta}_x \right\rangle \qquad (12)$$

*on each sample with perturbation $\hat{\delta}_x = \pm \varepsilon x$ only differ by a fixed multiple $\varepsilon^2$. Which is formally stated as*

$$\mathrm{Penalty_{LDAT}^2} = \varepsilon^2 \cdot \mathrm{Penalty_{IRM}}. \qquad (13)$$

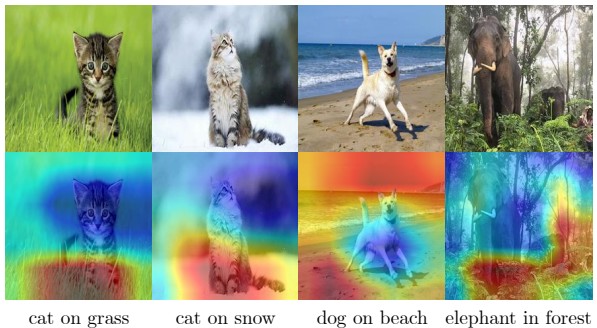

cat on grass    cat on snow    dog on beach    elephant in forest

Figure 2: Images in different domains from NICO dataset and corresponding attention heatmaps of a model trained by ERM. The model has a strong focus on background information.

Proofs of Proposition 1 and Remark 1 can be found in Appendix C.

## 4  UNDERSTANDING DOMAIN-WISE ADVERSARIAL TRAINING

The above connection between DAT and IRM highlights that our DAT is potentially helpful for addressing OOD problems. In this section, we further explore how domain-wise perturbations could help alleviate distribution shift in real-world scenarios. In particular, we notice that the domain-wise perturbations could successfully remove the domain-varying *background information*, which usually corresponds to the spurious features for image classification tasks. And we empirically verify this property by applying DAT to a well-designed OOD task based on background shift.

### 4.1  COMPARING ERM TO DAT FOR BACKGROUND REMOVAL

**ERM Learns Spurious Background Features**  Our understanding of DAT hinges on the insight that an image is composed of a foreground object and the corresponding background, and typically, the object is the invariant feature while the background is only spuriously correlated with the labels. However, models relying on the spurious background information will easily fail when encountering images from a different domain. This phenomenon is also empirically verified by Xiao et al. (2021), who find that models trained on an ImageNet-like dataset with ERM require image backgrounds for correctly classifying large portions of test sets. These findings point out the limitations of ERM and motivate us to find a solution that could effectively learn background-invariant classifier.

**Removing Background Information with DAT**  Comparing to the failure of ERM above, we notice that Domain-wise Adversarial Training (DAT) can be applied to eliminate the domain-wise background information with its domain-specific perturbations.

Take the NICO images in Figure 2 for an example, where samples from the environment "on grass" have a common background dominated by the green grass with low frequency, while the foreground object (*e.g.,* the cats) has complex and instance-specific shape and texture with much higher frequency. In fact, Moosavi-Dezfooli et al. (2017) show that a universal perturbation vector lies in a low dimensional subspace, which fits the background statistics and could be used to eliminate the low-frequency background factor. Therefore, when applying our DAT to these samples, the domain-wise perturbation will capture the common domain-specific background. And consequently, the domain-wise AT will help remove the dependence on these spurious background features.

### 4.2  EMPIRICAL VERIFICATION WITH CONTROLLED EXPERIMENTS

To verify the above analysis, we construct a synthetic OOD task to evaluate a classifier's dependence on background information. It is based on two datasets introduced by Xiao et al. (2021), Mixed-Same and Mixed-Rand, which are constructed from a subset of ImageNet images with the background of each image replaced by another background that is either from the same class (Mixed-Same) or a random class (Mixed-Rand). As they are perfect candidates for evaluating a classifier in terms of its background dependence, we construct a new OOD task by using Mixed-Same as the training domain and evaluating the learned classifier on Mixed-Rand as the test domain. If the clas-

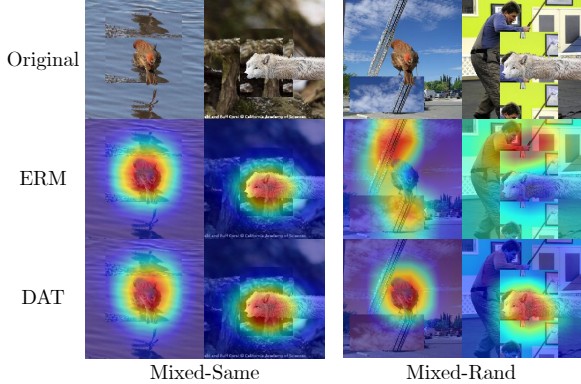

Figure 3: Images from Mixed-Same and Mixed-Rand and corresponding attention heatmaps of ERM and DAT, DAT effectively fixed the problem of excessive attention on uncorrelated background information.

sifier relies too much on background information, it will perform poorly in the test domain where objects and backgrounds are disentangled. In particular, experiment results show that ERM achieves a test accuracy of 71.9%, while DAT achieved 72.6% on the test domain with random background, which means that DAT has better generalization ability by effectively removing background information. Sample images from the dataset and corresponding attention heatmaps are shown in Figure 3, which demonstrate that ERM may lose its focus when background correlation is broken while DAT doesn't. Details for the experiment are shown in Appendix A.1

## 5 EXPERIMENTS

### 5.1 EXPERIMENTAL SETUP

For the experiments, we follow the setting in Ye et al. (2021) and evaluate OOD generalization on both the two types of distribution shift, diversity shift and correlation shift. In particular, we select four representative tasks. For correlation shift, we use CMNIST (Arjovsky et al., 2019), a synthetic dataset on digit classification which has color as the spurious feature, and NICO (He et al., 2020), a real-world dataset on object classification with context as the spurious feature. As for diversity shift, we use PACS (Li et al., 2017) and Terra Incognita (Beery et al., 2018), which are both real-world datasets with four domains. To ensure fair evaluation, we conduct all our experiments following the evaluation pipeline of DomainBed (Gulrajani & Lopez-Paz, 2021). Specifically, we use the same strategy for dataset splitting and model selection as in Ye et al. (2021) for each of these tasks. For datasets except for NICO, one of the domains is used as the test domain. We train the models in each run treating one of the domains as the test domain with the rest of the domains as training domains. Then report the average accuracy of all domains. For NICO, the training, test, evaluation, and test domains are predefined. We train the models on training domains, evaluate them on the evaluation domain for model selection, and report their accuracy on the test domain. More details of the experimental settings could be found in Appendix A.2.

When training models using DAT, we first perform standard data augmentation Gulrajani & Lopez-Paz (2021), then proceed with the update on perturbation and model parameters as shown in 1, where the perturbed samples are clipped to the legal range after data augmentation.

As discussed in Section 4, DAT can reduce the influence of background even when no domain labels are given, which corresponds to the single-source domain generalization setting. To see how this could help in this setting, we conduct experiments which strengthen this claim. Experimental setting and results can be found in Appendix B.

### 5.2 EVALUATION ON BENCHMARK DATASETS

We compare our results with previous works, including vanilla ERM, invariance-based methods including IRM (Ahuja et al., 2020), robust optimization methods including GroupDRO (Sagawa et al., 2020), distribution matching methods including MMD (Li et al., 2018b) and CORAL (Sun & Saenko, 2016), a method based on domain classifier (DANN, (Ganin et al., 2016)) and various other algorithms. The results for the CMNIST dataset are adopted from Gulrajani & Lopez-Paz

Table 1: Test accuracy (%) on four representative tasks for OOD Robustness. According to OOD-Bench (Ye et al., 2021), two are dominated by correlation shift, CMNIST and NICO, and two are dominated by diversity shift, PACS and TerraInc. We highlight the top two results on each task.

| Algorithm | Correlation shift | | Diversity shift | | Avg |
|---|---|---|---|---|---|
| | CMNIST | NICO | PACS | TerraInc | |
| ERM (Baseline) | $58.5 \pm 0.3$ | $71.4 \pm 1.3$ | $81.5 \pm 0.0$ | $42.6 \pm 0.9$ | 63.5 |
| VREx (Krueger et al., 2021) | $56.3 \pm 1.9$ | $71.0 \pm 1.3$ | $81.8 \pm 0.1$ | $40.7 \pm 0.7$ | 62.5 |
| GroupDRO (Sagawa et al., 2020) | $61.2 \pm 0.6$ | $\mathbf{71.8 \pm 0.8}$ | $80.4 \pm 0.3$ | $36.8 \pm 1.1$ | 62.6 |
| IRM (Ahuja et al., 2020) | $\mathbf{70.2 \pm 0.2}$ | $67.6 \pm 1.4$ | $81.1 \pm 0.3$ | $42.0 \pm 1.8$ | $\mathbf{65.2}$ |
| ARM (Zhang et al., 2020) | $63.2 \pm 0.7$ | $63.9 \pm 1.8$ | $81.0 \pm 0.4$ | $39.4 \pm 0.7$ | 61.9 |
| RSC (Huang et al., 2020) | $58.5 \pm 0.5$ | $69.7 \pm 0.3$ | $\mathbf{82.8 \pm 0.4}$ | $\mathbf{43.6 \pm 0.5}$ | 63.7 |
| DANN (Ganin et al., 2016) | $58.3 \pm 0.2$ | $68.6 \pm 1.1$ | $81.1 \pm 0.4$ | $39.5 \pm 0.2$ | 61.9 |
| MMD (Li et al., 2018b) | $63.4 \pm 0.7$ | $68.3 \pm 1.0$ | $81.7 \pm 0.2$ | $38.3 \pm 0.4$ | 62.9 |
| MTL (Blanchard et al., 2021) | $57.6 \pm 0.3$ | $70.2 \pm 0.6$ | $81.2 \pm 0.4$ | $38.9 \pm 0.6$ | 62.0 |
| MLDG (Li et al., 2018a) | $58.4 \pm 0.2$ | $51.6 \pm 6.1$ | $73.0 \pm 0.4$ | $27.3 \pm 2.0$ | 52.6 |
| SagNet (Nam et al., 2021) | $58.2 \pm 0.3$ | $69.3 \pm 1.0$ | $81.6 \pm 0.4$ | $42.3 \pm 0.7$ | 62.9 |
| CORAL (Sun & Saenko, 2016) | $57.6 \pm 0.5$ | $68.3 \pm 1.4$ | $81.6 \pm 0.6$ | $38.3 \pm 0.7$ | 61.5 |
| Mixup (Yan et al., 2020) | $58.4 \pm 0.2$ | $66.6 \pm 0.9$ | $79.8 \pm 0.6$ | $39.8 \pm 0.3$ | 61.2 |
| AT (sample-wise) (Goodfellow et al., 2014) | $57.9 \pm 0.4$ | $70.5 \pm 0.7$ | $\mathbf{82.0 \pm 0.2}$ | $42.6 \pm 0.3$ | 63.3 |
| UAT (Shafahi et al., 2020) | $58.7 \pm 2.3$ | $69.1 \pm 1.2$ | $80.7 \pm 0.4$ | $41.9 \pm 1.8$ | 62.6 |
| WRM (Sinha et al., 2018) | $57.9 \pm 3.3$ | $68.2 \pm 1.0$ | $80.4 \pm 0.0$ | $26.1 \pm 1.5$ | 58.2 |
| ADA (Volpi et al., 2018) | $56.3 \pm 0.4$ | $69.5 \pm 1.9$ | $80.2 \pm 0.2$ | $41.2 \pm 0.7$ | 61.8 |
| DAT (our work) | $\mathbf{68.4 \pm 2.0}$ | $72.6 \pm 1.7$ | $\mathbf{82.0 \pm 0.1}$ | $42.7 \pm 0.7$ | $\mathbf{66.4}$ |

(2021), which is the average of three domains, while the results of the other datasets are adopted from Ye et al. (2021). Apart from that, we implement and test four AT based algorithms including sample-wise adversarial training (AT (Goodfellow et al., 2014)), universal adversarial training (UAT (Shafahi et al., 2020) (which do not distinguish between different training domains), WRM (Sinha et al., 2018), and Adversarial Data Augmentation (ADA (Volpi et al., 2018)).

From Table 1, we observe that on a variety of OOD benchmark datasets, DAT has a performance consistently better than ERM. In particular, DAT achieves state-of-the-art performance under both diversity and correlation shift. In comparison, IRM only outperforms DAT on CMNIST and sample-wise AT fails on three of the four datasets. Compared to other AT-based algorithms (*e.g.,* sample-wise AT, UAT, WRM, and ADA), DAT has superior performance by considering a domain-wise perturbation that removes domain-varying spurious features. The results demonstrate DAT's effectiveness in dealing with domain discrepancy. RSC Huang et al. (2020) has the most superior performance under diversity shift, this may due to that RSC masks large gradients, which prevents the model from becoming over-confident by only capturing a few dominant

Table 2: Comparison of test accuracy of different perturbation radius $\varepsilon$ and step size $\alpha$ of $\ell_2$-norm bounded DAT on the NICO dataset.

| Radius $\varepsilon$ | Step Size $\alpha$ | Acc (%) |
|---|---|---|
| $[10^{-2}, 10^{-1}]$ | $[10^{-4}, 10^{-3}]$ | $72.6 \pm 1.7$ |
| | $[10^{-3}, 10^{-2}]$ | $72.0 \pm 2.1$ |
| | $[10^{-2}, 10^{-1}]$ | $68.9 \pm 1.5$ |
| $[10^{-1}, 10^{0}]$ | $[10^{-3}, 10^{-2}]$ | $71.2 \pm 0.4$ |
| | $[10^{-2}, 10^{-1}]$ | $66.6 \pm 1.7$ |
| | $[10^{-1}, 10^{0}]$ | $69.4 \pm 0.4$ |
| $[10^{0}, 10^{1}]$ | $[10^{-3}, 10^{-2}]$ | $64.6 \pm 0.3$ |
| | $[10^{-2}, 10^{-1}]$ | $67.9 \pm 2.0$ |
| | $[10^{-1}, 10^{0}]$ | $67.4 \pm 0.4$ |

inant features that are not invariant. However, RSC does not use domain labels, which makes it fail to consider the change of correlation between features and label class when generalizing to a new domain and leads to poor performance under correlation shift. In comparison, DAT solves the same problem in a reversed way, by perturbing inputs to reduce dominant features, and results in a "weaker" augmentation compared to RSC under diversity shift. But DAT considers domain difference explicitly, thus having better performance than RSC under correlation shift. We also note that DAT performs slightly worse than IRM on CMNIST, which could be due to the data generation process of CMNIST. As a synthetic dataset, CMNIST uses colored digits instead of colored background as the spurious features. In contrast, in real-world datasets like NICO (Figure 2), the spurious features mainly lie in the background instead of the object. This suggests that our method is more suitable for alleviating real-world distribution shifts.

## 5.3 ANALYSIS

We conduct experiments to understand better what our algorithm learns and how the magnitude of hyperparameters affects its performance.

**Qualitative Analysis through Semantic Graphs**  We use GradCam (Selvaraju et al., 2017; Gildenblat & contributors, 2021) to visualize the attention heatmaps of models trained by ERM, sample-wise AT, and DAT on the NICO dataset. The results are shown in Figure 4, where we can see that DAT puts more attention on the object itself instead of the strongly correlated background, while ERM and sample-wise AT tend to use environmental features instead.

**Perturbation Radius and Step Size**  We investigate the effect of perturbation radius $\varepsilon$ and perturbation step size $\alpha$ on the NICO dataset. The results are shown in Table 2. From the results, we find that the perturbation radius $\varepsilon$ has a huge impact on the OOD accuracy. When $\varepsilon$ is too large ($> 10^{-1}$), it begins to hurt the invariant feature and causes performance degradation (from 72.9% to 67.9%). The step size $\alpha$ has a smaller influence, and choosing a value about $1/100$ to $1/10$ times the size of $\varepsilon$ would be appropriate.

**Norm of Perturbations**  We also investigate the effect of the norm used in DAT. We perform experiments on the NICO dataset using $\ell_\infty$-norm bounded DAT. The results are shown in Table 3. Although there are slight differences, DAT bounded by both $p = 2$ and $p = \infty$ can achieve state-of-the-art performance (72.6% and 72.5%) on the NICO dataset.

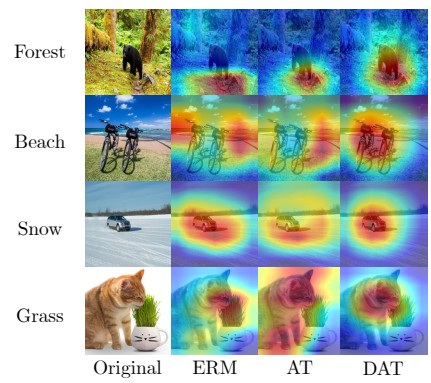

Figure 4: Attention heatmaps of ERM, (sample-wise) AT, and our DAT on the NICO dataset. Compared to ERM and AT, DAT has a more precise focus on the object itself.

## 6 DISCUSSIONS

**Comparison with Sample-wise AT**  Previous works (Yi et al., 2021; Hendrycks et al., 2021) try to exploit *sample-wise* AT as a data augmentation strategy to get higher OOD performance. However, the performance only improves when the distribution shift is dominated by diversity shift, *e.g.,* noise, and blurring. Otherwise, the performance might be degraded, as shown in Table 1. One possible explanation is that because sample-wise AT fails to capture the domain-level variations as DAT. As a result, it could possibly add perturbations to the invariant features and lead to inferior performance, especially under correlation shift.

Table 3: Test accuracy of $\ell_\infty$-norm bounded DAT on the NICO dataset.

| Radius $\varepsilon$ | Step Size $\alpha$ | Acc (%) |
|---|---|---|
| $[10^{-3}, 10^{-2}]$ | $[10^{-4}, 10^{-3}]$ | $68.5 \pm 0.2$ |
| $[10^{-2}, 10^{-1}]$ | $[10^{-3}, 10^{-2}]$ | $67.9 \pm 2.4$ |
| $[10^{-1}, 10^{0}]$ | $[10^{-3}, 10^{-2}]$ | $72.5 \pm 1.6$ |

**Comparison with Invariant Causal Prediction**  A thread of methods including ICP (Peters et al., 2016), IRM (Arjovsky et al., 2019), and IGA (Koyama & Yamaguchi, 2020) try to find the invariant data representations that could induce an invariant classifier. They have superior performance on synthetic datasets like CMNIST but fail to outperform ERM on real-world datasets (tasks dominated by both correlation shift and diversity shift). We believe their failures might be attributed to the lack of prior information in their invariant learning principles. While in our DAT, we effectively exploit the foreground-background difference in image classification tasks through domain-wise perturbations.

## 7 CONCLUSION

In this work, we carefully analyze the similarity between IRM and adversarial training in a domain-wise fashion and establish a formal connection between OOD and adversarial robustness. Based on this connection, we propose a new adversarial training method for domain generalization, namely Domain-wise Adversarial Training (DAT). We show that it could effectively remove the spurious background features in image classification and obtain superior performance on benchmark datasets. Notably, our DAT could outperform ERM consistently on tasks dominated by both correlation shift and diversity shift, while previous methods typically fail in one of the two cases.

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

## A  EXPERIMENT SETTINGS AND MORE DETAILED RESULTS

We train all the models based on the DomainBed benchmark presented in Gulrajani & Lopez-Paz (2021).

### A.1  MIXED-SAME AND MIXED-RAND

To construct the dataset, we use the validation set of Mixed-Same (4185 images) provided by Xiao et al. (2021) as the training environment and use the validation set of Mixed-Rand (4185 images) as the testing (OOD) environment.

For training, we use pretrained ResNet-18 as backbone and train it for 5000 iterations. The hyperparameters are same to those proposed in Gulrajani & Lopez-Paz (2021), except that the learning rate searching interval is altered to $[10^{-5.5}, 10^{-4.5}]$, while we set hyperparameters search space of DAT as $\varepsilon \in [10^{-0.5}, 10^0]$, $\alpha \in [10^{-1.5}, 10^{-1}]$.

### A.2  TYPICAL OOD DATASETS

We conduct experiments on four typical OOD datasets, including CMNIST, NICO, Terra Incognita, and PACS. We split the NICO dataset using the same strategy in Ye et al. (2021) for a fair comparison.

For network architecture, models trained on CMNIST adopt the four-layer convolution network MNIST-CNN provided in DomainBed benchmark (Gulrajani & Lopez-Paz, 2021), while other datasets use ResNet-18 as the backbone. For Terra Incognita and PACS, pretrained ResNet-18 is used. While on NICO, we use unpretrained ResNet-18 as the pretraining dataset contains photos that are largely overlapped with ImageNet classes.

For model selection, models trained on PACS and Terra Incognita are selected by training-domain validation, which selects the model with highest training accuracy averaged across all training domains. For NICO, an extra OOD validation set is used. CMNIST, due to the large correlation shift, uses test-domain validation as its model selection criteria.

DAT, AT, and UAT models are trained for an appropriate number of iterations to ensure convergence, that is, 5000 for Terra Incognita, 8000 for NICO, 10000 for CMNIST and PACS.

When training NICO using DAT, we clamp the single sample adversarial loss to $(0, 2)$ to obtain a better domain-wise perturbation as suggested in Shafahi et al. (2020).

We use the same hyperparameters as those proposed in Ye et al. (2021) whenever possible while setting the hyperparameters search space for DAT, sample-wise AT and UAT as in Table 6 and 5. We use DAT with perturbation bounded by $\ell_2$-norm in our experiments. For sample-wise AT, we use a 10-step $\ell_2$ PGD (Madry et al., 2018).

For WRM, we set the step number to 10 in the inner maximization, while searching for the learning rate in the inner maximization in the range $[10^{-2}, 10^{-1}]$ and $\gamma$ in the range $[10^{-0.5}, 10^{0.5}]$. We train the models for 5000 iterations to ensure convergence.

For ADA, we set the step number to 10 in the inner maximization, while searching for the adversarial learning rate in the range $[10^{-1}, 10^{0.5}]$, $\gamma$ in the range $[0.5, 1.5]$, the number of steps in the min-phase

in the range $[10^{0.5}, 10^{2.5}]$, and the number of whole adversarial phases in the range $[10^{0.5}, 10^2]$. We train the models for 5000 iterations to ensure convergence.

Table 4: Hyperparameter Search Space for DAT on Typical OOD Datasets.

| Dataset | Radius $\varepsilon$ | Step Size $\alpha$ |
|---------|---------|---------|
| CMNIST | $[10^{-1}, 10^2]$ | $[10^{-2}, 10^1]$ |
| NICO | $[10^{-2}, 10^{-1}]$ | $[10^{-4}, 10^{-3}]$ |
| PACS | $[10^{-1}, 10^2]$ | $[10^{-2}, 10^1]$ |
| TerraInc | $[10^{-0.5}, 10^{0.7}]$ | $[10^{-2}, 10^{-1}]$ |

Table 5: Hyperparameter Search Space for (sample-wise) AT on Typical OOD Datasets.

| Dataset | Radius $\varepsilon$ | Step Size $\alpha$ |
|---------|---------|---------|
| CMNIST | $[10^{-1}, 10^1]$ | $[10^{-2}, 10^0]$ |
| NICO | $[10^{-1}, 10^0]$ | $[10^{-2}, 10^{-1}]$ |
| PACS | $[10^{-2}, 10^{-1}]$ | $[10^{-3}, 10^{-2}]$ |
| TerraInc | $[10^{-1}, 10^0]$ | $[10^{-2}, 10^{-1}]$ |

Table 6: Hyperparameter Search Space for UAT on Typical OOD Datasets.

| Dataset | Radius $\varepsilon$ | Step Size $\alpha$ |
|---------|---------|---------|
| CMNIST | $[10^{-1}, 10^2]$ | $[10^{-2}, 10^1]$ |
| NICO | $[10^{-2}, 10^{-1}]$ | $[10^{-4}, 10^{-3}]$ |
| PACS | $[10^{-1}, 10^2]$ | $[10^{-2}, 10^1]$ |
| TerraInc | $[10^0, 10^{0.5}]$ | $[10^{-2}, 10^{-1}]$ |

The domain split results for ERM, UAT, and DAT are shown in Table 7, 8, and 9, where the results of ERM on CMNIST are from Gulrajani & Lopez-Paz (2021) and all other results come from our experiments.

Table 7: Test accuracy of ERM, UAT, and DAT on CMNIST.

(a) Training domain vaidation.

| Algorithm | 0.1 | 0.2 | 0.9 | Average |
|-----------|-----|-----|-----|---------|
| ERM | $\mathbf{72.7 \pm 0.2}$ | $73.2 \pm 0.3$ | $10.0 \pm 0.0$ | $52.0 \pm 0.1$ |
| UAT | $72.1 \pm 0.1$ | $\mathbf{73.7 \pm 0.1}$ | $10.0 \pm 0.1$ | $52.0 \pm 0.1$ |
| DAT | $72.4 \pm 0.2$ | $\mathbf{73.7 \pm 0.1}$ | $\mathbf{10.2 \pm 0.2}$ | $\mathbf{52.1 \pm 0.1}$ |

(b) Test domain vaidation (Oracle).

| Algorithm | 0.1 | 0.2 | 0.9 | Average |
|-----------|-----|-----|-----|---------|
| ERM | $72.3 \pm 0.6$ | $73.1 \pm 0.3$ | $30.0 \pm 0.3$ | $58.5 \pm 0.3$ |
| UAT | $75.3 \pm 6.4$ | $71.9 \pm 0.7$ | $29.0 \pm 0.2$ | $58.7 \pm 2.3$ |
| DAT | $\mathbf{78.5 \pm 4.8}$ | $\mathbf{79.3 \pm 0.2}$ | $\mathbf{47.5 \pm 2.6}$ | $\mathbf{68.4 \pm 2.0}$ |

Table 8: Test accuracy of ERM, UAT, and DAT on PACS.

(a) Training domain vaidation.

| Algorithm | Artpaint | Cartoon | Photo | Sketch | Average |
|---|---|---|---|---|---|
| ERM | $\mathbf{80.5 \pm 0.8}$ | $74.2 \pm 0.5$ | $\mathbf{94.7 \pm 0.5}$ | $72.9 \pm 2.1$ | $80.6 \pm 0.6$ |
| UAT | $79.9 \pm 0.6$ | $74.7 \pm 0.8$ | $92.6 \pm 0.4$ | $75.5 \pm 2.1$ | $80.7 \pm 0.4$ |
| DAT | $80.0 \pm 0.3$ | $\mathbf{77.6 \pm 0.8}$ | $92.6 \pm 0.9$ | $\mathbf{77.6 \pm 0.5}$ | $\mathbf{82.0 \pm 0.1}$ |

(b) Test domain vaidation (Oracle).

| Algorithm | Artpaint | Cartoon | Photo | Sketch | Average |
|---|---|---|---|---|---|
| ERM | $79.7 \pm 1.0$ | $\mathbf{76.9 \pm 0.3}$ | $\mathbf{94.2 \pm 0.4}$ | $78.3 \pm 1.5$ | $\mathbf{82.3 \pm 0.4}$ |
| UAT | $\mathbf{80.4 \pm 2.1}$ | $75.0 \pm 0.4$ | $93.5 \pm 0.5$ | $\mathbf{79.3 \pm 0.4}$ | $82.0 \pm 0.6$ |
| DAT | $79.7 \pm 0.1$ | $74.9 \pm 0.7$ | $93.9 \pm 0.6$ | $78.1 \pm 0.4$ | $81.7 \pm 0.2$ |

Table 9: Test accuracy of ERM, UAT, and DAT on TerraInc.

(a) Training domain vaidation.

| Algorithm | L100 | L38 | L43 | L46 | Average |
|---|---|---|---|---|---|
| ERM | $53.2 \pm 1.4$ | $31.9 \pm 0.3$ | $50.2 \pm 0.9$ | $\mathbf{33.6 \pm 0.3}$ | $42.2 \pm 0.4$ |
| UAT | $47.2 \pm 3.3$ | $\mathbf{36.9 \pm 1.6}$ | $\mathbf{50.3 \pm 0.3}$ | $32.8 \pm 0.3$ | $41.8 \pm 1.2$ |
| DAT | $\mathbf{53.8 \pm 1.6}$ | $35.1 \pm 2.3$ | $\mathbf{50.3 \pm 0.3}$ | $31.7 \pm 0.6$ | $\mathbf{42.7 \pm 0.7}$ |

(b) Test domain vaidation (Oracle).

| Algorithm | L100 | L38 | L43 | L46 | Average |
|---|---|---|---|---|---|
| ERM | $\mathbf{55.8 \pm 1.1}$ | $40.8 \pm 2.0$ | $49.8 \pm 1.3$ | $35.0 \pm 0.2$ | $45.3 \pm 0.3$ |
| UAT | $50.5 \pm 2.6$ | $40.3 \pm 0.2$ | $50.7 \pm 0.3$ | $34.3 \pm 0.5$ | $44.0 \pm 0.6$ |
| DAT | $54.1 \pm 2.3$ | $\mathbf{42.5 \pm 2.1}$ | $\mathbf{52.7 \pm 0.8}$ | $\mathbf{35.1 \pm 1.0}$ | $\mathbf{46.1 \pm 0.6}$ |

# B    ON SINGLE-DOMAIN DOMAIN GENERALIZATION

It can be seen that our proposed DAT can also work with a single domain. In this case, DAT reduces to UAT as a special case. In this section, we present a deeper understanding on the effectiveness of DAT on single-domain generalization, as well as the comparison between DAT and ensembled UAT. Specifically, in Section B.1, we perform experiments under single-source domain generalization setting, which show that DAT is still effective under this scenario by removing the common background in the domain. In Section B.2, we compare DAT with the ensemble of UAT, the results suggest that DAT has extra benefits by training a single model on multiple domains.

## B.1    SINGLE-SOURCE DOMAIN GENERALIZATION

To show that our method can also be applied to single-domain generalization scenarios, we conduct experiments with similar settings to Appendix A.2 on the four benchmark datasets, except that we train on one of the domains and report its test accuracy on all other domains. For the NICO dataset, as it comes with a validation domain, we train the models on one of the training domains, using the validation domain for model selection, then report its accuracy on the test domain.

We train all models using ERM/DAT for 5000 iterations to ensure convergence and set the hyper-parameter searching space of DAT as shown in Table 10. All other settings are the same as those shown in Appendix A.2.

Table 10: Hyperparameter searching space for DAT on typical OOD datasets (single-source domain generalization)

| Dataset | Radius $\varepsilon$ | Step Size $\alpha$ |
|---------|----------------------|--------------------|
| CMNIST | $[10^{-1}, 10^2]$ | $[10^{-2}, 10^1]$ |
| NICO | $[10^{-2}, 10^{-1}]$ | $[10^{-4}, 10^{-3}]$ |
| PACS | $[10^{-2}, 10^{-1}]$ | $[10^{-3}, 10^{-2}]$ |
| TerraInc | $[10^{-0.5}, 10^{0.7}]$ | $[10^{-2}, 10^{-1}]$ |

We list the domain-average results in Table 11.

Table 11: Average test accuracy of ERM, DAT on four representative tasks (single-source domain generalization)

| Algorithm | CMNIST | NICO | PACS | TerraInc | Avg |
|-----------|--------|------|------|----------|-----|
| ERM | 45.8 | 63.2 | 59.8 | 27.3 | 49.0 |
| DAT | **46.0** | **64.4** | **59.9** | **28.1** | **49.6** |

We can see that our DAT enjoys superior performance on all four datasets compared to ERM, either under correlation shift or diversity shift. Although the difference is not as significant as in the multiple domain setting, it shows that our DAT works for both single-domain and multi-domain generalization scenarios. In particular, its advantages are more significant with multiple domains, where the domain-wise perturbation mechanism is more effective.

The domain split results for ERM and DAT under single-source domain generalization setting are shown in Table 12a to 12h.

## B.2 COMPARISON WITH THE ENSEMBLE OF UAT

To show that models trained using DAT on multi-domains are not a trivial ensemble of models trained on single domains, we train voting classifiers on CMNIST and NICO using UAT. The models are trained separately on each domain and then perform an equal weight voting to get prediction results. The results are shown in Table 13.

Table 13: Results of Ensemble UAT and DAT on CMNIST and NICO

| Algorithm | CMNIST | NICO |
|-----------|--------|------|
| Ensemble UAT | $58.2 \pm 2.3$ | $60.8 \pm 0.2$ |
| DAT | $\mathbf{68.4 \pm 2.0}$ | $\mathbf{72.6 \pm 1.7}$ |

From the results, we can see that invariance learned by DAT indeed performs much better than a trivial ensemble of independent UAT models. We believe that training DAT on multiple domains can help extract the domain-invariant features through sharing the same model across domains while removing domain-varying features with domain-specific perturbations. Spurious background information from multiple domains can help the classifier to identify the truly domain-invariant features. In comparison, single-domain UAT only has access to a single domain and can only learn invariance of a single domain, which leads to inferior performance.

## C PROOFS

We give a proof of Remark 1 first, then use a similar technique to prove Proposition 1.

Table 12: Test accuracy (single-source domain generalization)

(a) ERM on CMNIST

| Training Domain | 0.1 | 0.2 | 0.9 | Avg |
|---|---|---|---|---|
| 0.1 | \ | $79.8 \pm 0.2$ | $21.2 \pm 0.2$ | 50.5 |
| 0.2 | $88.7 \pm 0.4$ | \ | $35.2 \pm 1.3$ | **62.0** |
| 0.9 | $21.0 \pm 0.9$ | $29.1 \pm 0.7$ | \ | **25.1** |

(b) DAT on CMNIST

| Training Domain | 0.1 | 0.2 | 0.9 | Avg |
|---|---|---|---|---|
| 0.1 | \ | $80.0 \pm 0.1$ | $27.9 \pm 4.7$ | **54.0** |
| 0.2 | $90.0 \pm 0.1$ | \ | $32.4 \pm 1.0$ | 61.2 |
| 0.9 | $18.3 \pm 1.1$ | $27.1 \pm 1.1$ | \ | 22.7 |

(c) ERM on NICO

| Training Domain | Test Accuracy |
|---|---|
| 1 | $56.8 \pm 3.5$ |
| 2 | $69.6 \pm 2.0$ |
| Avg | 63.2 |

(d) DAT on NICO

| Training Domain | Test Accuracy |
|---|---|
| 1 | $61.7 \pm 0.9$ |
| 2 | $67.1 \pm 0.3$ |
| Avg | **64.4** |

(e) Test accuracy of ERM on PACS (single-source domain generalization)

| Training Domain | Artpaint | Cartoon | Photo | Sketch | Avg |
|---|---|---|---|---|---|
| Artpaint | \ | $64.9 \pm 0.7$ | $94.2 \pm 0.1$ | $61.5 \pm 4.1$ | **73.5** |
| Cartoon | $61.9 \pm 2.7$ | \ | $76.6 \pm 1.6$ | $69.5 \pm 1.1$ | 69.3 |
| Photo | $66.9 \pm 0.3$ | $26.9 \pm 0.1$ | \ | $35.5 \pm 0.7$ | **43.1** |
| Sketch | $47.3 \pm 0.1$ | $65.7 \pm 0.9$ | $46.6 \pm 0.8$ | \ | 53.2 |

(f) Test accuracy of DAT on PACS (single-source domain generalization)

| Training Domain | Artpaint | Cartoon | Photo | Sketch | Avg |
|---|---|---|---|---|---|
| Artpaint | \ | $59.6 \pm 0.0$ | $94.2 \pm 0.8$ | $48.7 \pm 2.3$ | 67.5 |
| Cartoon | $68.1 \pm 0.1$ | \ | $83.9 \pm 1.0$ | $69.1 \pm 1.3$ | **73.7** |
| Photo | $64.5 \pm 1.4$ | $27.6 \pm 5.8$ | \ | $30.9 \pm 4.5$ | 41.0 |
| Sketch | $52.2 \pm 0.2$ | $63.4 \pm 1.4$ | $56.4 \pm 1.1$ | \ | **57.3** |

(g) Test accuracy of ERM on TerraInc (single-source domain generalization)

| Training Domain | L100 | L38 | L43 | L46 | Avg |
|---|---|---|---|---|---|
| L100 | \ | $24.9 \pm 4.1$ | $26.8 \pm 0.5$ | $27.1 \pm 4.5$ | **26.3** |
| L38 | $42.3 \pm 0.6$ | \ | $12.5 \pm 1.2$ | $13.2 \pm 1.5$ | 22.7 |
| L43 | $34.2 \pm 9.8$ | $31.9 \pm 0.9$ | \ | $30.1 \pm 0.5$ | 32.1 |
| L46 | $24.9 \pm 6.2$ | $13.0 \pm 2.8$ | $46.6 \pm 2.0$ | \ | 28.3 |

(h) DAT on TerraInc

| Training Domain | L100 | L38 | L43 | L46 | Avg |
|---|---|---|---|---|---|
| L100 | \ | $23.3 \pm 1.4$ | $24.3 \pm 0.7$ | $16.7 \pm 1.2$ | 21.4 |
| L38 | $44.6 \pm 7.4$ | \ | $17.3 \pm 1.0$ | $16.3 \pm 0.4$ | **26.1** |
| L43 | $35.2 \pm 7.4$ | $32.0 \pm 7.6$ | \ | $32.9 \pm 1.5$ | **33.4** |
| L46 | $24.3 \pm 4.0$ | $23.2 \pm 0.2$ | $47.5 \pm 0.2$ | \ | **31.7** |

**Remark 1 (Equivalence under Single-sample Environments)** *When the environments degenerate into a single data point, we have the following relationship: If $\varepsilon$ is sufficiently small, then for $\beta \cdot \Phi$ as a deep network with any activation function, the penalty term of IRMv1 (Eq. 4) on each sample and the square of the maximization term of Linearized version of DAT (LDAT, obtained by first-order approximation of DAT)*

$$\text{Penalty}_{\text{LDAT}} = \left\langle \nabla_x \ell \left( \beta^T \Phi(x), y \right), \pm \hat{\delta}_x \right\rangle \tag{14}$$

*on each sample with $\hat{\delta}_x = \pm \varepsilon x$ only differ by a fixed multiple $\varepsilon^2$. Which is formally stated as*

$$\text{Penalty}_{\text{LDAT}}^2 = \varepsilon^2 \cdot \text{Penalty}_{\text{IRM}}. \tag{15}$$

**Proof of Remark 1**   Note that we presume $\Phi$ is piece-wise linear. We represent the output logit of the given deep network as $\beta^\top \Phi(x) = \beta^\top \Phi_x x$ where $\Phi_x$ is the matrix related to sample $x$ due to the fact of different activation patterns.

We first derive the form of $\text{Penalty}_{\text{LDAT}}$ by first-order approximation of Eq. 9.

$$\ell(\beta^\top \Phi(x + \delta_e), y) - \ell(\beta^\top \Phi(x), y) \approx \left\langle \nabla_x \ell(\beta^\top \Phi(x), y), \delta_e \right\rangle \tag{16}$$

Letting $\delta_e = \pm \varepsilon x$, we have that

$$\text{Penalty}_{\text{LDAT}} = \left\langle \nabla_x \ell \left( \beta^T \Phi(x), y \right), \pm \hat{\delta}_x \right\rangle \tag{17}$$

The penalty term of IRMv1 on each sample is as follows:

$$
\begin{aligned}
& \left\| \nabla_{w|w=1.0} - \log(\sigma(w \cdot (y\beta^\top \Phi(x)))) \right\|^2 \\
&= \left\| -\frac{\sigma'(w \cdot (y\beta^\top \Phi(x)))}{\sigma(w \cdot (y\beta^\top \Phi(x)))} y\beta^\top \Phi(x) \right\|^2 \\
&= \left\| -(1 - \sigma(y\beta^\top \Phi(x))) y\beta^\top \Phi(x) \right\|^2 \\
&= (1 - \sigma(y\beta^\top \Phi(x)))^2 \left\| \beta^\top \Phi_x x \right\|^2
\end{aligned} \tag{18}
$$

For LDAT, note that the gradient *w.r.t.* $x$ is equal to

$$\nabla_x [-\log(\sigma(y\beta^\top \Phi(x)))] = -(1 - \sigma(y\beta^\top \Phi(x))) y\beta^\top \Phi_x$$

So the square of the penalty term of LDAT on each sample with perturbation $\pm \varepsilon x$ is:

$$[\langle \nabla_x - \log(\sigma(y\beta^\top \Phi(x))), \pm \varepsilon x \rangle]^2 = \varepsilon^2 (1 - \sigma(\beta^\top \Phi(x)))^2 \left\| \beta^\top \Phi_x x \right\|^2 \tag{19}$$

which is identical to Eq. 4 with a difference of multiple $\varepsilon^2$.

$$\square$$

**Proposition 1** *Consider each $D_e$ as the corresponding distribution of a particular training domain $e$. For any $\beta \cdot \Phi$ as a deep network with any activation function, the penalty term of IRMv1,* $\text{Penalty}_{\text{IRM}}$ *(Eq. 4), could be expressed as the square of a reweighted version of the penalty term of the above approximate target,* $\text{Penalty}_{\text{DAT}}$*(Eq. 10), on each environment $e$ with coefficients related to the distribution $D_e$, which could be stated as follows:*

$$\text{Penalty}_{\text{IRM}} = \left\| \mathbb{E}_{D_e}[L_x x] \right\|^2, \text{ and } \text{Penalty}_{\text{DAT}} = \left\| \mathbb{E}_{D_e} L_x \right\| \tag{20}$$

*where $L_x = (1 - \sigma(y\beta^\top \Phi_x x)) y\beta^\top \Phi_x$.*

**Proof of Proposition 1**   For IRMv1, from the proof of Remark 1 we know

$$\nabla_{w|w=1.0} \ell(w \cdot (\beta \cdot \Phi), y) = -(1 - \sigma(y\beta^\top \Phi_x x)) y\beta^\top \Phi_x x \tag{21}$$

So that $\text{Penalty}_{\text{IRM}}$ (suppose derivation and integration are commutable)

$$\left\| \nabla_{w|w=1.0} R^e (w \cdot (\beta \cdot \Phi)) \right\|^2 = \left\| \mathbb{E}_{D_e} (1 - \sigma(y\beta^\top \Phi_x x)) y\beta^\top \Phi_x x \right\|^2 \tag{22}$$

For DAT, on each environment

$$
\begin{aligned}
&\max_{\|\delta_e\|\leq\varepsilon} \mathbb{E}_{D_e}\left[\ell(\beta^\top\Phi(x+\delta_e),y)-\ell(\beta^\top\Phi(x),y))\,\right]\\
&\approx \max_{\|\delta_e\|\leq\varepsilon} \mathbb{E}_{D_e}\left[\left\langle\nabla_x\ell(\beta^\top\Phi(x),y),\delta_e\right\rangle\right]\\
&= \max_{\|\delta_e\|\leq\varepsilon} \left\langle\mathbb{E}_{D_e}\left[\nabla_x\ell(\beta^\top\Phi(x),y)\right],\delta_e\right\rangle\\
&=\varepsilon\left\|\mathbb{E}_{D_e}\left[\nabla_x\ell(\beta^\top\Phi(x),y)\right]\right\|
\end{aligned}
\tag{23}
$$

So $\mathrm{Penalty_{DAT}}$ is

$$
\left\|\mathbb{E}_{D_e}\left[\nabla_x\ell(\beta^\top\Phi(x),y)\right]\right\| = \left\|\mathbb{E}_{D_e}\left[-(1-\sigma(y\beta^\top\Phi(x)))y\beta^\top\Phi_x\right]\right\|
\tag{24}
$$

So we know $\mathrm{Penalty_{IRM}}$ can be regarded as the square of a *reweighted* version of $\mathrm{Penalty_{DAT}}$ with coefficient on each sample inside the expectation $x$.

$\square$

