# OpenReview forum: "Domain-wise Adversarial Training for Out-of-Distribution Generalization"
_ICLR.cc/2022/Conference — ICLR 2022 Submitted_

### Official Review · Reviewer_LGzX · 2021-10-27

**Correctness:** 3
**Technical Novelty And Significance:** 2
**Empirical Novelty And Significance:** 2
**Recommendation:** 6
**Confidence:** 5

**Details Of Ethics Concerns:**

No ethical concerns to report.

**Main Review:**

# Strengths
- The paper is (in my knowledge) one of the first to study the connections between methods for adversarial robustness and OOD generalization, whereas previous work has studied the connection between robustness vs in-domain accuracy, and OOD generalization vs in-domain accuracy.
- The structure and writing is clear (apart from minor grammatical errors) and the main idea of the paper is well stated.

# Weaknesses
1. The theoretical derivations in Section 3 assume that multiple environments are available for training.  The proposed method (DAT) also relies on this assumption.  While this may be a useful assumption for settings in which multiple environments are indeed available (eg. PACS), it does not hold for benchmarks such as "DIGITS" (train on MNIST, test on 4 other datasets) where only a single environment is available (aka single-source domain generalization).  On the other hand, standard adversarial training makes no such assumption. It would be useful if the authors can shed light on whether the connection between IRM and DAT exists for single-source domain generalization.
2. The introduction and abstract mentions domain-invariant features and distribution shift/OOD generalization.  But the paper only deals with the special case of "multi-source" or multi-environment domain generalization.  It is not clear whether DAT (and IRM) would also work for single-source domain generalization.  The theory and empirical results in this paper do not provide any insights into this.  I would recommend making this distinction clear.
3. The presentation of the experiments and results could be improved substantially.  Critical details about the datasets, which environment/domains are used for training/testing are missing.  This makes it hard to interpret the results.
4. Details about other baselines from Table 1 are missing (what type of training do they use, do some of these fall under AT/IRM/ERM?). Why is RSC the best model under diversity-shift? What is the difference between RSC and DAT that could be the reason?
5. There is little analysis of why DAT is better than IRM on some domains and worse on others.  While the visualizations in Figure 2 and 3 are useful, they are anecdotal and do not provide analysis at scale.

# Questions
- From Algorithm 1 it seems that N domains , N different values of $\delta_e$ are updated and stored (one for each domain).  One could also think of adversarial training separately on each domain, thereby obtaining N different values of $\theta$. These can be ensembled to get a prediction.  How does DAT compare with such an "ensembling" of AT applied separately to each domain?
- From Table 1, it appears that sample-wise AT is worse than ERM for all 4 benchmarks
    - On CMNIST:  AT < ERM < DAT < IRM
    - On NICO:  IRM < AT < ERM < DAT
    - On PACS:  AT < IRM < ERM < DAT
    - On TerraInc:  IRM < ERM~=AT < DAT

    This means that while the while the comparative relation between IRM and DAT is somewhat clear, that between IRM and AT is mixed.
    What is the insight here?  Why is IRM better than AT in some case, and worse in others?
-  It is stated that colored backgrounds are more realistic than colored digits.  Why is it so?  Why is it more "realistic" to have spurious correlations in the background?  For instance, the SVHN digits dataset contains digits with both colored backgrounds and colored content (digits).  Perhaps this additional benchmark could be used to substantiate the statements that hypothesize why DAT < IRM for CMNIST.

# Feedback
- Footnote 1 is very important, and with the growing number of papers on adversarial robustness, ood generalization, domain shift, etc., it is prudent to point out conflicting/equivalent terminology, as is done by the authors here.
- Page 1. Paragraph 1. "carefully designed perturbations" -- I would advise the authors to distinguish between
    1. perturbations optimized for each image (i.e. using gradient descent like in Madry et al.)
    2. universal adversarial perturbations (Moosavi-Dezfooli et al. CVPR 2017 https://arxiv.org/abs/1610.08401 among others) and
    3. "common corruptions/perturbations" (Hendrycks et al. ICLR 2019 https://arxiv.org/abs/1903.12261)
Perhaps it would be useful to state which of these are designed, and which of these are simply learnt.
- To add to Strengths #1, although a principled study has not been done to connect robustness and OOD generalization, there are some papers that do use adversarial training (adversarial data augmentation) to improve OOD performance or some aspects of distribution shift. For example:
    1. Volpi et al. NIPS 2018, https://arxiv.org/abs/1805.12018
    2. Qiao et al. CVPR 2020, https://arxiv.org/abs/2003.13216
    3. Gokhale et al. AAAI 2021, https://arxiv.org/abs/2003.13216
- Section 3 uses $\beta$, $\phi$ for the classifier, while Algorithm 1 uses $\theta$ -- please make this consistent.
- It would be better to show the heatmaps from DAT in Figure 1 (similar to Figure 2).


# Minor
## Terminology
- This is probably pedantic, but in many places the terminology "diversity-shift task" or "correlation-shift task" is used.  I believe that this should be changed to "performance under diversity shift" or "image classification task under diversity shift", since *shift is not a task*, but rather a situation for evaluating robustness/generalization.

## Grammar / Typos
- Page 1. "are proposed" --> "have been proposed", "IRM attracts significant attention" --> "IRM has attracted signification attention"
- Page 1. Para 3. "promising" --> promise
- many instances of subject-verb agreement errors. please run a grammar check when updating this version.

# Review Summary
The paper introduces DAT -- a version of adversarial training specifically designed for training datasets containing multiple environments.  Connections between DAT, AT, IRM are derived formally.  While the motivation and theory behind the paper is sound, empirical results are not conclusive and the paper is lacking in terms of analysis and interpretation of results.  Additionally, it is not clear if DAT would also apply for broader cases of domain generalization where there may not be "known" environments (i.e. domain labels are not available) or only a single environment/domain is available for training.

During discussions with authors, more empirical evidence was provided and most of my questions were addressed.  I am leaning towards acceptance.


**Summary Of The Paper:**

The paper investigates whether adversarial training (AT) could be used for extracting domain-invariant features, and whether AT can benefit OOD generalization.  The paper first shows the relationship between AT and IRM (invariant risk minimization).  A new version of AT is proposed -- called "Domain-wise Adversarial Training (DAT)"; with IRM being a version of DAT.  Results suggest that DAT can outperform ERM under correlation-shift and diversity shift.

**Summary Of The Review:**

Well motivated idea that derives connections between IRM and AT; experiments are found lacking.
During discussions with authors, more empirical evidence was provided and most of my questions were addressed.

---

> ### Author Response · Authors · 2021-11-20
> **Response to Reviewer LGzX (4/4)**
>
> **Q7**: The comparative relation between IRM and AT is mixed in Table 1. What is the insight here? Why is IRM better than AT in some cases, and worse in others?
>
> **A7**: As discussed in **Section 6**, sample-wise AT has better performance under diversity shift by improving distributional robustness, as is shown in [1-3], but it fails to capture the domain-level variations, which leads to inferior performance under correlation shift. As for IRM, it aims to find a classifier that is invariant among training domains, which is suitable for Colored MNIST dataset that is artificially designed in the IRM paper [4] and has a large correlation shift between training and test domains. However, IRM fails in real-world datasets, which might be attributed to the lack of prior information in their invariant learning principles.
>
> Besides, we note that due to the limit of time, we did not tune AT exhaustively at the time of submission. We have updated our results of AT on PACS leads to the following results.
>
> | Algorithm | CMNIST           | NICO             | PACS             | TerraInc         | Avg      |
> | --------- | ---------------- | ---------------- | ---------------- | ---------------- | -------- |
> | IRM       | **70.2 +/- 0.2** | 67.6 +/- 1.4     | 81.1 +/- 0.3     | 42.0 +/- 1.8     | **65.2** |
> | AT        | 57.9 +/- 0.4     | **70.5 +/- 0.7** | **82.0 +/- 0.2** | **42.6 +/- 0.3** | 62.8     |
>
> From the results, we can see that AT performs better than IRM on all real-world datasets (NICO, PACS, TerraInc) but has inferior performance on CMNIST. We believe this is due to that CMNIST is an artificially designed dataset that has a great discrepancy measured in $\ell_p$-norm between samples with different colors, which makes it difficult for AT to eliminate the influence of color and leads to bad results as choosing a large perturbation hurts invariant features as well.
>
> [1] Certifying Some Distributional Robustness with Principled Adversarial Training, https://arxiv.org/abs/1710.10571
>
> [2] Learning to Learn Single Domain Generalization, https://arxiv.org/abs/2003.13216
>
> [3] Improved OOD Generalization via Adversarial Training and Pre-training, https://arxiv.org/abs/2105.11144
>
> [4] Invariant Risk Minimization, https://arxiv.org/abs/1907.02893
>
> ---
>
> **Q8**: Why is it more "realistic" to have spurious correlations in the background?
>
> **A8**: As discussed in **Section 4.1**, this observation is verified by several empirical works. In particular,  [1, 2] suggest that for a large number of natural image datasets, models can attain non-trivial test accuracy on datasets consisting of only backgrounds, which means that there exists a strong correlation between background and front scene objects on these datasets. Although SVHN may be a good candidate to test the effect of varying image color, it is a dataset with a single domain, which is non-trivial for DAT to adapt to.
>
> [1] Noise or Signal: The Role of Image Backgrounds in Object Recognition, https://arxiv.org/abs/2006.09994
>
> [2] Object Recognition with and without Objects, https://arxiv.org/abs/1611.06596
>
> ---
>
> **Q9**: The terminology "diversity-shift task" or "correlation-shift task" is not accurate.
>
> **A9**: We have changed the terminology to "under diversity/correlation shift" in the revision.
>
> ---
>
> **Q10**: "carefully designed perturbations" on **Page 1 Paragraph 2** is ambiguous.
>
> **A10**: We have changed "carefully designed perturbations" to "perturbations optimized for each image" as you suggested.
>
> ---
>
> **Q11**: More related works are required on the study of the connection between robustness and OOD generalization.
>
> **A11**: We have added reference to the following papers to **Section 2** following your suggestion.
>
> [1] Generalizing to Unseen Domains via Adversarial Data Augmentation, https://arxiv.org/abs/1805.12018
>
> [2] Learning to Learn Single Domain Generalization, https://arxiv.org/abs/2003.13216
>
> [3] Attribute-Guided Adversarial Training for Robustness to Natural Perturbations, https://arxiv.org/abs/2012.01806
>
> ---
>
> **Q12**: The symbol used for the classifier is not consistent in **Section 3** and **Algorithm 1**.
>
> **A12**: We have changed $\theta$ to $\beta$ and $\phi$ in **Algorithm 1** to ensure consistency with **Section 3**.
>
> ---
>
> Thanks again for your detailed review. We hope our extended results and explanations could address your concerns. Please let us know if you have any additional questions.

---

> ### Author Response · Authors · 2021-11-20
> **Response to Reviewer LGzX (3/4)**
>
> **Q2**: Does the connection between IRM and DAT exist for single-source domain generalization? Why DAT requires multiple domains while standard AT does not have this requirement?
>
> **A2**: From our formulation of **Prop 1**, we can see that $Penalty_{IRM}
>     = \left\\|E_{D_{e}} [L_x x]\right\\|^2 $ and $ Penalty_{DAT} = \left\\|E_{D_{e}} L_x\right\\|$ (Eq.11) are both defined for environment $e$, which still exists when only one of the environments are used as training domain. As for the comparison to standard AT, we want to emphasize that DAT differs from AT by generating adversarial examples for a whole domain instead of a single sample, which requires domain label to work and leads to superior performance on multi-source domain generalization as shown in **Table 1**.
>
> | Algorithm | CMNIST           | NICO             | PACS             | TerraInc         | Avg      |
> | --------- | ---------------- | ---------------- | ---------------- | ---------------- | -------- |
> | ERM       | 58.5 +/- 0.3     | 71.4 +/- 1.3     | 81.5 +/- 0.0     | 42.6 +/- 0.9     | 63.5     |
> | AT        | 57.9 +/- 0.4     | 70.5 +/- 0.7     | **82.0 +/- 0.2** | 42.6 +/- 0.3     | 62.8     |
> | DAT       | **68.4 +/- 2.0** | **72.6 +/- 1.7** | **82.0 +/- 0.1** | **42.7 +/- 0.7** | **66.4** |
>
> ---
>
> **Q3**: More details about the datasets and baselines are required.
>
> **A3**: As you suggested, in the revision, we have added descriptions on the baselines and datasets to **Section 5.1**, including a brief introduction of the baselines and datasets and a more detailed discussion on the evaluation scheme.
>
> ---
>
> **Q4**: Why is RSC the best model under diversity-shift? What is the difference between RSC and DAT that could be the reason?
>
> **A4**: RSC [1] masks large gradients, which prevents the model from becoming over-confident by only capturing a few dominant features that are not invariant. This helps RSC to deal with diversity shift, for example, the change of texture in PACS, and therefore leads to better performance. However, RSC does not use domain labels, which makes it fail to consider the change of correlation between features and label class when generalizing to a new domain and leads to poor performance under correlation shift. In comparison, DAT solves the same problem in a reversed way, by perturbing inputs to reduce dominant features, and results in a "weaker" augmentation compared to RSC under diversity shift. But DAT considers domain difference explicitly, thus having better performance than RSC under correlation shift.
>
> [1] Self-Challenging Improves Cross-Domain Generalization, https://arxiv.org/abs/2007.02454
>
> ---
>
> **Q5**: More analyses are required on why DAT is better than IRM on some domains and worse on others.
>
> **A5**: We note that we have included a discussion on why DAT performs worse than IRM on CMNIST dataset in **Section 5.2.** We believe that it is mainly because CMNIST is an artificial dataset with no background information, so the performance of DAT (designed to remove the spurious features in the background) is limited in this dataset. Instead, on more real-world datasets, like NICO, PACS, and TerraInc, they have very informative background information varying across domains. In these scenarios, DAT is more effective than IRM and IRM even fails behind ERM. This shows that a certain degree of priors (as in DAT) needed to be injected for real-world datasets.
>
> ---
>
> **Q6**: What is the difference between a model trained using DAT and an ensemble of models trained using UAT on independent domains?
>
> **A6**: The key difference is that training DAT on multiple domains can help 1) extract the domain-invariant features through sharing the same model across domains, 2) while removing domain-varying features with domain-specific perturbations. Spurious background information from multiple domains can help the classifier to identify the truly domain-invariant features. In comparison, single-domain UAT only has access to a single domain and can only learn invariance to a single domain. Therefore, an ensemble of independent single-domain UAT models cannot achieve invariance to multiple domains.
>
> Empirically, we also verify this analysis on our benchmark datasets. Specifically, we get 58.2 +/- 2.3 (UAT ensemble) vs **68.4 +/- 2.0** (DAT) on CMNIST and 60.8 +/- 0.2 (UAT ensemble) vs **72.6 +/- 1.7** (DAT) on NICO. This shows that invariance learned by DAT indeed performs much better than a trivial ensemble of independent UAT models.

---

> > ### Comment · Reviewer_LGzX · 2021-11-22
> > **Satisfied with Q2,Q3,Q4,Q6.  Not entirely convinced about Q5+Q7+Q8**
> >
> > Thank you for the detailed answers, authors.  I'm happy with the empirical results for Q6 (UAT ensemble vs DAT).
> > You may consider adding some of the insights that you've mentioned under Q4 and Q6 in the main paper.
> >
> > I'm adding a combined response to your response 3/4 and 4/4 to my review.
> >
> > - About Q5 (and the related questions in Q7, Q8) -- I think your observations based on empirical results on CMNIST make sense.
> > - However I don't believe that spurious background correlations are more realistic than other types of domain shift.  Definitely these types of datasets are great testbeds for controlled study on the effect of backgrounds on classifier performance, but that doesn't mean that they are more "realistic". One could argue that changes in lighting conditions, noise, weather artifacts (rain/fog/snow/..), changing camera positions/zoom/angles etc are more realistic since these will be encountered for a camera deployed in the real world on an everyday basis.
> >
> > Overall, I appreciate that you could provide empirical evidence to answer some of my questions.  This has helped me understand the efficacy of the method better.  Hope that my other minor comments on terminology/references etc were useful.

---

> > > ### Author Response · Authors · 2021-11-22
> > > **Thanks for your feedbacks**
> > >
> > > Thanks for your valuable suggestions and for appreciating our feedbacks.
> > > Following your suggestions, in the revision, we have added
> > >
> > > 1) RSC’s results to **Section 5.2**;
> > > 2) the single-domain generalization results to **Appendix B.1**; and
> > > 3) the comparison with the ensemble of UAT to **Appendix B.2**.
> > >
> > > As for your concern on whether spurious correlations in image background are more “realistic”, we do agree that the word “realistic” may cause ambiguity. As discussed in **Section 4.1**, because we adopt a domain-wise perturbation for all samples in the domain, our method is more suitable for **low-frequency features** that are common across the entire domain. Often, these features are shown as background information, such as the sky and the woods, as shown in the NICO dataset. In other scenarios, as you have mentioned, it could also be lighting and weather that also affect the foreground. In either case, these real-world spurious features often to be relatively steady in the same domain compared to the objects to be classified. Therefore, rigorously speaking, the inherit prior of our method is not about "spurious background correlations are more realistic", but instead, "domain-wise steady features are more realistic".
> > >
> > > We hope this explanation could address your concerns. Please let us know if there is anything that we could/should further clarify.

---

> > > > ### Comment · Reviewer_LGzX · 2021-11-24
> > > > **Most concerns addressed. Suggest rethinking "realistic"**
> > > >
> > > > Thanks authors.  I think most of my concerns have been addressed, and I am inclined to recommend this paper for acceptance.  I am okay with the change that you have suggested
> > > > `"domain-wise steady features are more realistic".`

---

> ### Author Response · Authors · 2021-11-20
> **Response to Reviewer LGzX (2/4)**
>
> For completeness, we also show the detailed results per domain as follows. We can see that in most cases, our DAT indeed performs better than ERM.
>
> ### 1) CMNIST
>
> ERM Average: 45.8%
>
> | Training Domain | 0.1          | 0.2          | 0.9          | Avg      |
> | --------------- | ------------ | ------------ | ------------ | -------- |
> | 0.1             | \            | 79.8 +/- 0.2 | 21.2 +/- 0.2 | 50.5     |
> | 0.2             | 88.7 +/- 0.4 | \            | 35.2 +/- 1.3 | **62.0** |
> | 0.9             | 21.0 +/- 0.9 | 29.1 +/- 0.7 | \            | 25.1     |
>
> DAT Average: **46.0%**
>
> | Training Domain | 0.1          | 0.2          | 0.9          | Avg      |
> | --------------- | ------------ | ------------ | ------------ | -------- |
> | 0.1             | \            | 80.0 +/- 0.1 | 27.9 +/- 4.7 | **54.0** |
> | 0.2             | 90.0 +/- 0.1 | \            | 32.4 +/- 1.0 | 61.2     |
> | 0.9             | 18.3 +/- 1.1 | 27.1 +/- 1.1 | \            | **22.7** |
>
> ### 2) NICO
>
> ERM Average: 63.2%
>
> | Training Domain | Test Accuracy |
> | --------------- | ------------- |
> | 1               | 56.8 +/- 3.5  |
> | 2               | 69.6 +/- 2.0  |
> | Avg             | 63.2          |
>
> DAT Average: **64.4%**
>
> | Training Domain | Test Accuracy |
> | --------------- | ------------- |
> | 1               | 61.7 +/- 0.9  |
> | 2               | 67.1 +/- 0.3  |
> | Avg             | **64.4**      |
>
> As the NICO dataset comes with a validation domain, we train the models on one of the training domains, using validation domain for model selection, then report its accuracy on test domain.
>
> ### 3) PACS
>
> ERM Average: 59.8%
>
> | Training Domain | Artpaint     | Cartoon      | Photo        | Sketch       | Avg      |
> | --------------- | ------------ | ------------ | ------------ | ------------ | -------- |
> | Artpaint        | \            | 64.9 +/- 0.7 | 94.2 +/- 0.1 | 61.5 +/- 4.1 | **73.5** |
> | Cartoon         | 61.9 +/- 2.7 | \            | 76.6 +/- 1.6 | 69.5 +/- 1.1 | 69.3     |
> | Photo           | 66.9 +/- 0.3 | 26.9 +/- 0.1 | \            | 35.5 +/- 0.7 | **43.1** |
> | Sketch          | 47.3 +/- 0.1 | 65.7 +/- 0.9 | 46.6 +/- 0.8 | \            | 53.2     |
>
> DAT Average: **59.9%**
>
> | Training Domain | Artpaint     | Cartoon      | Photo        | Sketch       | Avg      |
> | --------------- | ------------ | ------------ | ------------ | ------------ | -------- |
> | Artpaint        | \            | 59.6 +/- 0.0 | 94.2 +/- 0.8 | 48.7 +/- 2.3 | 67.5     |
> | Cartoon         | 68.1 +/- 0.1 | \            | 83.9 +/- 1.0 | 69.1 +/- 1.3 | **73.7** |
> | Photo           | 64.5 +/- 1.4 | 27.6 +/- 5.8 | \            | 30.9 +/- 4.5 | 41.0     |
> | Sketch          | 52.2 +/- 0.2 | 63.4 +/- 1.4 | 56.4 +/- 1.1 | \            | **57.3** |
>
> ### 4) TerraInc
>
> ERM Average: 27.3%
>
> | Training Domain | L100         | L38          | L43          | L46          | Avg      |
> | --------------- | ------------ | ------------ | ------------ | ------------ | -------- |
> | L100            | \            | 24.9 +/- 4.1 | 26.8 +/- 0.5 | 27.1 +/- 4.5 | **26.3** |
> | L38             | 42.3 +/- 0.6 | \            | 12.5 +/- 1.2 | 13.2 +/- 1.5 | 22.7     |
> | L43             | 34.2 +/- 9.8 | 31.9 +/- 0.9 | \            | 30.1 +/- 0.5 | 32.1     |
> | L46             | 24.9 +/- 6.2 | 13.0 +/- 2.8 | 46.6 +/- 2.0 | \            | 28.3     |
>
> DAT Average: **28.1%**
>
> | Training Domain | L100         | L38          | L43          | L46          | Avg      |
> | --------------- | ------------ | ------------ | ------------ | ------------ | -------- |
> | L100            | \            | 23.3 +/- 1.4 | 24.3 +/- 0.7 | 16.7 +/- 1.2 | 21.4     |
> | L38             | 44.6 +/- 7.4 | \            | 17.3 +/- 1.0 | 16.3 +/- 0.4 | **26.1** |
> | L43             | 35.2 +/- 7.4 | 32.0 +/- 7.6 | \            | 32.9 +/- 1.5 | **33.4** |
> | L46             | 24.3 +/- 4.0 | 23.2 +/- 0.2 | 47.5 +/- 0.2 | \            | **31.7** |

---

> > ### Comment · Reviewer_LGzX · 2021-11-22
> > **Response to results on Single-Source DG**
> >
> > Thanks for providing these additional results.  Indeed there seem to be improvements on multiple datasets. Clearly DAT is better than ERM even for the SSDG setting.  Of course, since SSDG is significantly harder than multi-source (as multiple environments are not available, the challenge here is more data augmentation than invariance) -- different types of solutions might be needed.
> >
> > These SSDG results could be added to Appendix.

---

> ### Author Response · Authors · 2021-11-20
> **Response to Reviewer LGzX (1/4)**
>
> Thanks for your valuable feedback. We address them in details as follows.
>
> ---
>
> **Q1**: How would DAT perform in the single-environment domain generalization setting?
>
> **A1**: We note that our method mainly focuses on the multi-domain generalization setting as studied in the related works [1-3]. Nevertheless, as you mentioned, our method can also be applied to single-domain generalization. We compare our DAT with ERM on the four benchmark datasets in **Table 1** in the single-domain setting. We list the domain-average results in the following Table.
>
>
> | Method | ColorMNIST | NICO      | PACS      | TerraInc  | Avg       |
> | ------ | ---------- | --------- | --------- | --------- | --------- |
> | ERM    | 45.8%      | 63.2%     | 59.8%     | 27.3%     | 49.0%     |
> | DAT    | **46.0%**  | **64.4%** | **59.9%** | **28.1%** | **49.6%** |
>
> We can see that our DAT enjoys superior performance on all four datasets, either correlation-shift tasks or diversity-shift tasks.
>
> Although the difference is not as significant as in the multiple domain setting, it shows that our DAT works for both single-domain and multi-domain generalization scenarios. In particular, its advantages are more significant with multiple domains, where the domain-wise perturbation mechanism is more effective.
>
> [1] Invariant Risk Minimization, https://arxiv.org/abs/1907.02893
>
> [2] In Search of Lost Domain Generalization, https://arxiv.org/abs/2007.01434
>
> [3] OoD-Bench: Benchmarking and Understanding Out-of-Distribution Generalization Datasets and Algorithms, https://arxiv.org/abs/2106.03721

---

### Official Review · Reviewer_Gexh · 2021-11-02

**Correctness:** 4
**Technical Novelty And Significance:** 3
**Empirical Novelty And Significance:** Not applicable
**Recommendation:** 5
**Confidence:** 4

**Main Review:**

**Strengths**

- The idea of performing universal adversarial training at domain level for domain generalization is new and interesting. It is reasonable to assume each domains will be biased towards specific nuisances that result in biased classifiers, and I believe that tackling those at domain level instead of dataset or sample level is promising. Results suggest the efficacy of this approach.

- The paper is overall well written and pleasant to read.

- The GradCAM analysis (Figure 3) is very effective, and supports the paper's claims. I would encourage the Authors to include more of those in the supplementary material - they are very informative.

**Weaknesses**

- *Experimental analysis.* The numbers in Table 1 are not from (Gulrajani and Lopez-Paz, ICLR 2021). Such work provides a fair way of reporting consistent, statistically significant results across the literature, and it has been widely accepted by the community. If there is any reason not to use the baselines from this work, it should be explained. Also, more difficult datasets are missing - the most important being DomainBed. Related to Table 1, some bold numbers are not really superior to baseline results in a statistically significant fashion - for example, on NICO and Terra Incognita overall superiority of the proposed method over ERM cannot be claimed.

- *Missing baselines/related work.* (Shankar et al. ICLR 2018) and (Volpi et al. NeurIPS 2018), which are the two main works exploring adversarial training for domain generalization (multi-source and single source, respectively) are not cited. These are the two main references that address AT for DG, and given the nature of the proposed method they should be included in the experimental analysis.

- *Relationship between DAR and IRM.* I cannot understand the connection provided in Section 3.3. While I see the similarity between the two regularizers (Sec. Eq. 10 and 8) - the two are fundamentally different: the DAT one penalizes large gradients wrt to the input, whereas the IRM one penalizes large gradients wrt the classifier parameters. I cannot understand how the two derivatives are compared in Eq. 11 and 13. Can the Authors comment more on this? On a different and more subjective note, I do not believe that an analysis based on a single-sample domain assumption if informative for the paper (Remark 1).

**Clarity/writing:**

- The description of correlation-shift and diversity-shift in my opinion is not clear: a figure and/or a concrete example would help.

- A better reference for AT - when introduced - would be (Goodfellow et al. ICLR 2015), which actually propose it. (Madry et al., ICLR 2018) propose methods to do it more effectively.

- (Gilmer et al., ICML 2019) explore the connection between AT and OOD robustness, opening several questions; (Taori et al., NeurIPS 2020) discuss the impact of AT to OOD robustness in a variety of conditions. These work should be at least mentioned.

- When introducing DRO in the contenxt of AT/DG, (Sinha et al., ICLR 2018) and (Volpi et al., NeurIPS 2018) better fit as references in my opinion, since they indeed propose to use AT for DRO (the former for adversarial robustness, the latter for DG).

- I am assuming that also standard data augmentation is performed, as suggested in (Gulrajani and Lopez-Paz, ICLR 2021). Does it happen after Step 1 in Algorithm 1? This should be clarified.

**Summary Of The Paper:**

This paper focuses on domain generalization (DG). It proposes a method based on adversarial training (AT): the main idea is learning universal adversarial perturbations at domain level (domain adversarial perturbations, DAT), and rely on those within a standard AT routine. The relation between the proposed DAT and invariant risk minimization (IRM) is provided, and the method is empirically tested on standard domain generalization benchmarks.

**Summary Of The Review:**

I think the idea of treating every domain independently while performing AT for DG is very interesting. It is indeed reasonable to assume that each domain will rely on different nuisances that are not relevant for the task at hand, and removing those at domain level seem to be indeed effective.

Yet, I'm not fully convinced by the experimental analysis and the connection with IRM (in order of importance, in my assessment). (Gulrajani and Lopez-Paz, ICLR 2021) provide clear guidelines to compare DG methods in a fair fashion, and this benchmark has been widely accepted in the community (cf. recent works on DG). Since this benchmark is embraced in this submission, I do not understand why Table 1 i) provides worse performing baselines and ii) does not include results related to all the benchmarks.

I look forward reading the Author response and participating in the follow-up discussions.

---

> ### Author Response · Authors · 2021-11-20
> **Response to Reviewer Gexh (2/2)**
>
> **Q4**: Missing baselines/related works.
>
> **A4**: Thanks. We have added [1] and [2] to related works in **Section 2**, and include their results in **Table 1 in Section 5.2**. The results are shown below.
>
> | Algorithm | CMNIST           | NICO             | PACS             | TerraInc     | Avg      |
> | --------- | ---------------- | ---------------- | ---------------- | ------------ | -------- |
> | ERM       | 58.5 +/- 0.3     | 71.4 +/- 1.3     | 81.5 +/- 0.0     | 42.6 +/- 0.9 | 63.5     |
> | AT        | 57.9 +/- 0.4     | 70.5 +/- 0.7     | **82.0 +/- 0.2** | 42.6 +/- 0.3 | 62.8     |
> | WRM [1]   | 57.9 +/- 3.3     | 68.2 +/- 1.0     | 80.4 +/- 0.0     | 26.1 +/- 1.5 | 58.2     |
> | ADA [2]   | 56.3 +/- 0.4     | 69.5 +/- 1.9     | 80.2 +/- 0.2     | 41.2 +/- 0.7 | 61.8     |
> | DAT       | **68.4 +/- 2.0** | **72.6 +/- 1.7** | **82.0 +/- 0.1** | 42.7 +/- 0.7 | **66.4** |
>
> From the table, we can see that DAT has superior performance than other AT-based algorithms, which demonstrated its effectiveness on dealing with domain discrepancy, which other algorithms fail to consider.
>
> [1] Certifying Some Distributional Robustness with Principled Adversarial Training, https://arxiv.org/abs/1710.10571
>
> [2] Generalizing to Unseen Domains via Adversarial Data Augmentation, https://arxiv.org/abs/1805.12018
>
> ---
>
> **Q5**: How to interpret the results in Eq.11 and Eq.13?
>
> **A5**: $Penalty_{IRM}
>     = \left\\|E_{D_{e}} [L_x x]\right\\|^2 $ and
>    $ Penalty_{DAT} = \left\\|E_{D_{e}} L_x\right\\|$   (Eq.11)
>
> The expressions above (Eq.11) show that both penalty of DAT and penalty of IRM on an environment $e$ can be expressed using the expectation of $L_x$ on the distribution of $e$, where the penalty of IRM has an extra reweighting term $x$ inside the expectation. This connects the two kinds of penalty together and thus shows the similarity between IRM and DAT.
> When considering an oversimplified case of Eq.11 where each environment consists of a single data point, we can get Eq.13 as follows:
>
> $Penalty^2_{LDAT} = \varepsilon^{2}  Penalty_{IRM}$   (Eq.13)
>
> where $Penalty_{LDAT} = \left\langle\nabla_{x} \ell\left(\beta^{T} \Phi(x), y\right), \pm\hat\delta_x\right\rangle$ is the penalty of LDAT, a first-order approximation of DAT with its perturbation chosen as $\hat\delta_x = \pm\varepsilon x$, a multiple of the input. This shows that in this scenario the penalty of IRM and the square of the penalty of LDAT only differs by a fixed multiple.
>
> ---
>
> **Q6**: More detailed discussions are required on correlation-shift and diversity-shift.
>
> **A6**: Thanks for your advice. We have added a figure (**Figure 1**) and a more detailed discussion on the two kinds of shifts in **Section 2**.
>
> ---
>
> **Q7**: When is standard data augmentation performed in the pipeline?
>
> **A7**: Indeed, as you suggested, the data augmentation step is performed after Step 1 in **Algorithm 1**. In each iteration, the algorithm first takes in the input and augment it following the same augmentation strategy in [1], then generates adversarial examples and updates network parameters based on the augmented input. We have added a more detailed description of the pipeline to **Section 5.1**.
>
> [1] In Search of Lost Domain Generalization, https://arxiv.org/abs/2007.01434
>
> ---
>
> Thanks for your constructive comments. Please let us know if you have additional questions.

---

> ### Author Response · Authors · 2021-11-20
> **Response to Reviewer Gexh (1/2)**
>
> Thank you for your positive comments! We address your main concerns as follows.
>
> ---
>
> **Q1**: OOD Bench is used instead of DomainBed as benchmark and results on some datasets are missing.
>
> **A1**: We use the procedure proposed in OOD bench [1] to evaluate our algorithm as it identifies the two kinds of shifts that exist in common OOD generalization tasks. The benchmark they proposed based on these two kinds of shifts fully presents the domain generalization ability of algorithms, so we used it instead of DomainBed [2]. As for the results on other datasets, just as stated in the paper, our algorithm has the property of suppressing the influence of background on classification results, which makes it more suitable for datasets with background information. OfficeHome and CelebA used in [1], however, do not fit this criterion well. For example, in OfficeHome, samples in both the Clipart and Product domains have no background at all; while for CelebA, it is **an artificially created dataset** where the domains are partitioned using face-related attributes, so there is no natural change in the background across domains. As a result, CelebA is not suitable for our DAT method which is designed to remove domain-varying background information. Indeed, our DAT achieves inferior performance on these two datasets, with an accuracy of 85.9 (DAT) vs **87.2** (ERM) on CelebA and 61.1 (DAT) vs **63.3** (ERM) on OfficeHome. This shows that our DAT indeed relies on the prior that natural domain variations mostly happen in the background.
>
> [1] OoD-Bench: Benchmarking and Understanding Out-of-Distribution Generalization Datasets and Algorithms, https://arxiv.org/abs/2106.03721
>
> [2] In Search of Lost Domain Generalization, https://arxiv.org/abs/2007.01434
>
> ---
>
> **Q2**: Some of the results in **Table 1** are lower than those provided in [2].
>
> **A2**: Results of baselines in **Table 1** on PACS and Terra Incognita datasets are from [1] which are lower than results in [2] as ResNet-18 is used in [1] as backbone instead of ResNet-50 in [2] which could still serve as an accurate indicator of domain generalization performance.
>
> [1] OoD-Bench: Benchmarking and Understanding Out-of-Distribution Generalization Datasets and Algorithms, https://arxiv.org/abs/2106.03721
>
> [2] In Search of Lost Domain Generalization, https://arxiv.org/abs/2007.01434
>
> ---
>
> **Q3**: A better reference on the connection between AT and OOD robustness is required.
>
> **A3**: We have added the following reference to **Section 2** based on your suggestions.
>
> [1] Adversarial Examples Are a Natural Consequence of Test Error in Noise, https://arxiv.org/abs/1901.10513

---

> > ### Comment · Reviewer_Gexh · 2021-11-23
> > **Thank you for your response**
> >
> > Thank you for addressing my questions and extending the paper in these directions.
> >
> > Re: Q1 -- while I understand that specific methods perform well in specific conditions, if the proposed strategy is proposed as a "domain generalization" one, it should be tested on standard benchmarks. We need at least to assess that it does not compromise performance on other datasets. If the method improved performance on all datasets with varying background and maintained a comparable performance on all datasets without background differences, I would still consider it an effective one. The fact that it leads to lower performance on some datasets is worrisome (why is it compromising performance? Is it erasing important visual features?), and readers should be able to assess the overall performance. That is the positive aspect of DomainBed -- the global picture it provides.
> >
> > To summarize my point, all DomainBed results should be reported, also the ones where ERM out-performs the proposed method. Naturally, NICO results can still be reported, even though they are not part of DomainBed. Yet I believe it is very important that - as a community - we use the same benchmarks and compare with the same baselines, hence I think that the choice of not using the DomainBed baselines is sub-optimal in regard to fostering comparability with published and future works.
> >
> > I would consider this a valuable method if improved performance on datasets with background bias would not come at the price of lower performance on datasets without such bias. Or, if very specific and realistic applications exist where such background-biased datasets must be used. At this stage, not being convinced about the overall performance of the method, I cannot recommend acceptance, even though the comparison with standard AT methods makes the proposed strategy more promising.

---

> > > ### Author Response · Authors · 2021-11-29
> > > **Further Response to Reviewer Gexh**
> > >
> > >
> > > Thanks for appreciating our response. We will further elaborate our idea and add extensive results to address your concerns.
> > >
> > > ---
> > > **Differences between OOD-Bench [1] and DomainBed [2].** Here, we further elaborate on why we adopt OOD-Bench [1] as the benchmark by noting the differences between the two benchmarks. First, OOD-Bench **identifies both correlation shift and diversity shift, and encourage algorithms to address both cases**. In particular, they adopt test domain validation for correlation-shift tasks and training domain validation for tasks dominated by diversity shift. In comparison, the default setting of  DomainBed [2] only considers training domain validation, which could hardly tell apart the effectiveness of algorithms under correlation shift. Apart from that, most DomainBed datasets are diversity shift and **lack evaluation on real-world correlation shift tasks** (e.g. NICO). Third, OOD-Bench adopts a light-weighted backbone ResNet-18 that suits for fast evaluation. That being said, the two still share many common datasets, e.g., CMNIST, PACS, TerraInc, OfficeHome, etc.
> > >
> > > **Extended experiments on DomainBed.** Meanwhile, we agree with you on the importance of evaluation on common benchmarks. Albeit OOD-Bench is a promising new one, we would also like to provide DomainBed results for completeness and fair comparison following your suggestions. We note that due to the limit of time, we could only provide some preliminary results on medium-sized datasets, and those results might be further improved in the future with additional hyperparameter tuning. We list all the results related to DomainBed datasets in the following table.
> > >
> > > | Dataset | CMNIST (new)                 | NICO* (old) | RMNIST (new)     | VLCS  (new)           | PACS* (old)           | PACS  (new)            | TerraInc* (old)
> > > | --------- | ---------------- | ---------------- | ---| ---------------- | ---------------- | ----------------- | ---------------- |
> > > | Type | C | C | D| D | D | D | D| D|D|D|
> > > | ERM       | 51.5 ± 0.1   | 71.4  ± 1.3    | **98.0 ± 0.0**| **77.5 ± 0.4** | 81.5 ± 0.0     | 85.5 ± 0.2      | 42.6 ± 0.9     | 67.5 ± 0.5 | 41.2 ± 0.2 |
> > > | DAT       | **52.1 ± 0.1**| **72.6 ± 1.7**   | 97.9 ± 0.0   | 77.1 ± 0.2     | **82.0 ± 0.1** | **86.0  ± 0.4** | **42.7 ± 0.7** |
> > >
> > > **Note on experiment setup.** Two types of data: Correlation-shift (C) v.s. Diversity-shift (D). Experiments are conducted with DomainBed setting unless specified. Those marked with * are conducted with ResNet-18 for efficiency (as in OOD-Bench). The results for ERM are adopted from DomainBed [2,3]. Due to the limit of time, the results on the two large datasets in DomainBed, OfficeHome and DomainNet, will be provided in the future.
> > >
> > > **Analysis**.  From the results, we can see that DAT indeed performs consistently better in correlation-shift tasks. As for diversity-shift tasks, DAT also performs
> > > better than ERM on most of the datasets while having comparable performance on other datasets. This further justifies the effectiveness of DAT on addressing both kinds of distribution shift.
> > >
> > > [1] OoD-Bench: Benchmarking and Understanding Out-of-Distribution Generalization Datasets and Algorithms, https://arxiv.org/abs/2106.03721
> > >
> > > [2] In Search of Lost Domain Generalization, https://arxiv.org/abs/2007.01434
> > >
> > > [3] https://github.com/facebookresearch/DomainBed
> > >
> > > ---
> > > Hope this could address your concerns and we are looking forward to hearing from you.

---

### Official Review · Reviewer_HCbA · 2021-11-02

**Correctness:** 4
**Technical Novelty And Significance:** 2
**Empirical Novelty And Significance:** 2
**Recommendation:** 6
**Confidence:** 4

**Main Review:**

Strengths:
- The paper is well written and explores the challenging Domain Generalization settings.
- The paper highlights a relationship between IRM and AT
- Empirical results show that the method might be better than ERM for Domain Generalization settings.

Weaknesses:
Novelty.
- DAT is a trivial adaptation of UAT to DG settings.
- Furthermore, UAT is not compared to DAT in the experiments.
Empirical results.
- The advantage of the proposed DAT algorithm on diversity shift tasks is minimal compared to ERM.
- The aggregated performance improvements are mostly due to improvements in the CMNIST task. CMNIST is a synthetic digits dataset whose practical interest could be regarded as inferior compared to images datasets.
- Lack of results on domain splits. These could have been useful to get an high level insight on which scenarios the method is working best.
- Related to the above comment: there is no discussion on failure cases. Despite improving over ERM, it is hard to believe that the proposed method is improving in every split. A discussion on the potential downsides could have been as valuable as the method itself.

**Summary Of The Paper:**

The paper explores the relationship between Invariant Risk Minimization and Adversarial Training and proposes Domain-wise Adversarial Training, an adaptation of UAT designed specifically as a data augmentation layer for Domain Generalization settings.
Experiments on four benchmark datasets suggest that the proposed approach might be beneficial for DG when target domains suffer correlation/diversity shifts.

**Summary Of The Review:**

Although the inquired relationship between IRM and AT is interesting, the proposed approach is only marginally novel, as it is a simple adaptation of UAT to Domain Generalization settings. UAT acts as data augmentation, so it is not surprising that it is beneficial in DG settings.

Furthermore, I am not convinced by experimental results: there is only a limited insight on favorable scenarios, as domain split results are not presented, and improvements are mostly due do very good performance on a single dataset consisting of synthetic digits.

Still I am looking forward for the authors' response, and I hope that they can address my concerns (see Weaknesses).

---

> ### Author Response · Authors · 2021-11-20
> **Response to Reviewer HCbA**
>
> Thank you for your recognition of the clarity of this paper! Here is our answer to your concern.
>
> ---
>
> **Q1**: DAT is a trivial adaptation of UAT to DG settings and UAT should be compared to DAT in experiments.
>
> **A1**: Our proposed DAT is driven by the connection between IRM and adversarial training as discussed in **Section 3**, and we further verify its effectiveness in removing background information in **Section 4**. In this section, we tried to make DAT more general and principled, instead of involving too many hand-engineered complex tricks. Extensive experiments on real-world datasets show that, as a simple and general technique, DAT has superior performance to ERM under both correlation shift and diversity shift.
>
> For a more clear comparison between DAT and UAT, in the revision, we have evaluated them in benchmark datasets and the results are added to **Table 1**.
>
> | Algorithm | CMNIST           | NICO             | PACS             | TerraInc         | Avg      |
> | --------- | ---------------- | ---------------- | ---------------- | ---------------- | -------- |
> | ERM       | 58.5 +/- 0.3     | 71.4 +/- 1.3     | 81.5 +/- 0.0     | 42.6 +/- 0.9     | 63.5     |
> | UAT       | 58.7 +/- 2.3     | 69.1 +/- 1.2     | 80.7 +/- 0.4     | 41.8 +/- 1.2     | 62.6     |
> | DAT       | **68.4 +/- 2.0** | **72.6 +/- 1.7** | **82.0 +/- 0.1** | **42.7 +/- 0.7** | **66.4** |
>
> We can see that only DAT outperforms ERM consistently on the benchmark datasets (under either correlation shift or diversity shift), while UAT often falls behind ERM. This further strengthens our point that DAT is not a trivial adaptation of UAT.
>
> ---
>
> **Q2**: Minimal improvement of DAT over baselines under diversity shift.
>
> **A2**: From **Table 1 in Section 5.2** we know that most domain generalization algorithms at the moment cannot surpass ERM on tasks dominated by diversity shifts. Although RSC has great performance on these tasks, it performs much worse under correlation shifts, as shown in **Table 1**. DAT, however, outperforms ERM consistently and could account for both kinds of shifts.
>
> | Algorithm | CMNIST           | NICO             | PACS             | TerraInc         | Avg      |
> | --------- | ---------------- | ---------------- | ---------------- | ---------------- | -------- |
> | ERM       | 58.5 +/- 0.3     | 71.4 +/- 1.3     | 81.5 +/- 0.0     | 42.6 +/- 0.9     | 63.5     |
> | RSC       | 58.5 +/- 0.5     | 69.7 +/- 0.3     | **82.8 +/- 0.4** | **43.6 +/- 0.5** | 63.7     |
> | DAT       | **68.4 +/- 2.0** | **72.6 +/- 1.7** | 82.0 +/- 0.1     | 42.7 +/- 0.7     | **66.4** |
>
> ---
>
> **Q3**: Performance improvements are mainly due to CMNIST dataset, which is synthetic and has less importance.
>
> **A3**: Related to the above reply, as shown in **Table 1**, DAT performs consistently better than ERM on both kinds of tasks, which demonstrates its superior performance on OOD generalization, especially on real-world datasets including Mixed (**Section 4.2**) and NICO (**Section 5.2**). **Even without CMNIST**, DAT performs better than other baselines.
>
> ---
>
> **Q4**: Where are the results on domain splits and the discussion on potential downsides of DAT?
>
> **A4**: We indeed included the domain-split results in **Table 7, 8, 9 of Appendix A.2**. In the revision of the paper, we have added a more detailed description of the domain-split results and included the domain splits results of ERM and UAT for comparison. For example, we show the domain-split results on PACS dataset as below:
>
> | Training Domain | Artpaint         | Cartoon          | Photo            | Sketch           | Avg              |
> | --------------- | ---------------- | ---------------- | ---------------- | ---------------- | ---------------- |
> | ERM             | **80.5 +/- 0.8** | 74.2 +/- 0.5     | **94.7 +/- 0.5** | 72.9 +/- 2.1     | 80.6 +/- 0.6     |
> | DAT             | 80.0 +/- 0.3     | **77.6 +/- 0.8** | 92.6 +/- 0.9     | **77.6 +/- 0.5** | **82.0 +/- 0.1** |
>
> From the results above, we can see that DAT on PACS has a worse performance compared to ERM when the Photo domain is used as the test domain (92.6 (DAT) v.s. **94.7** (ERM)). We believe this is because the Photo domain contains rich background information. DAT works better when the Photo domain is used in training compared to other algorithms which exploit spurious background information while having inferior performance when the Photo domain is not used, as DAT may attack other useful features. This is consistent with our claim that DAT works by removing the background information.
>
> ---
>
> Thanks for your constructive comments. Hope our feedback could help answer your concerns!

---

> > ### Comment · Reviewer_HCbA · 2021-11-23
> > **Response**
> >
> > Thank you for your replies. I still find that the method doesn't show substantial improvements compared to ERM in most tasks (CMNIST dominates the average).
> > However, A1 addressed one of my main concerns about this work, therefore I've raised the score to marginally above acceptance treshold.

---

### Official Review · Reviewer_Ywim · 2021-11-02

**Correctness:** 3
**Technical Novelty And Significance:** 4
**Empirical Novelty And Significance:** 3
**Recommendation:** 8
**Confidence:** 3

**Main Review:**

Strengths:
- The proposed method is novel and an interesting modification/mix of the existing per-sample adversarial training and universal adversarial training (UAT).
- The connection between IRM and the proposed method is interesting, and the derivation/exposition of the connection is elegant
- The paper is well written, and the method is well presented
- The approach outperforms ERM and prior methods on several OOD benchmarks

Weaknesses:
- The authors claim that adversarial training can handle diversity shift well, but experiments in Table 1 do not support this (AT is worse than ERM in most benchmarks). It is also not easily decipherable if other adversarial methods listed in Table 1 could verify this claim (it might be good to indicate adversarial methods).
- Given that DAT is a “mix” between AT and UAT, I was missing a comparison to UAT in Table 1
- The comparison in Table 1 spans only a subset of the datasets used in either Ye et al. (2021) or (Gulrajani & Lopez-Paz, 2021). It would be good to know if the method performs less well on the other tasks or the rationale to limit the comparison to the chosen datasets.
- I found the experiment in Section 4.2 sounded interesting, but I could not find any results other than Figure 2. Though having some visualization is nice, a quantitative evaluation would be important there.
- Although I found the connection to IRM interesting, I was missing insights into why/how the differences in both methods could explain the benefit of DAT. For example, Proposition 1 suggests that IRM is an instance-reweighed version of DAT. It is unclear what the implications of this are on the performance of DAT compared to IRM.

**Summary Of The Paper:**

The paper tackles the problem of out-of-distribution (OOD) generalization of deep learning models with a novel adversarial training formulation (DAT) that introduces a shared adversarial perturbation per training domain.
The paper further establishes a connection between DAT, Invariant Risk Minimization (IRM), and standard adversarial training (AT).
The authors posit that DAT inherits the advantages of IRM (which is effective for correlation shift while it performs more poorly on diversity shift) and AT, which improves upon Empirical Risk Minimization (ERM) for diversity shift.
Experiments on several OOD benchmark datasets show that DAT outperforms prior works and ERM on average.

**Summary Of The Review:**

The paper tackles a vital problem (OOD generalization) with a novel adversarial training approach. The paper also illustrates an interesting connection to Invariant Risk Minimization. The method is supported by good performance on four OOD benchmark datasets, where it outperforms both IRM and standard adversarial training. Additional insights into the benefits of DAT compared to IRM (given the similarity) and a more extensive quantitative evaluation would strengthen the paper.

---

> ### Author Response · Authors · 2021-11-20
> **Response to Reviewer Ywim (2/2)**
>
> **Q4**: In **Section 4.2**, the quantitative results on Mixed dataset are missing.
>
> **A4**: Following your suggestions, we have added the quantitative results on the Mixed dataset to the last paragraph of **Section 4.2** as follows:
>
> > Experiment results show ERM achieves a test accuracy of 71.9%, while DAT achieves **72.6%** on the test domain with random background, which means that DAT has better generalization ability by effectively removing background information.
>
> ---
>
> **Q5**: Why is it possible for DAT to perform better than IRM?
>
> **A5**: DAT is different from IRM by directly penalizing the gradient w.r.t. the input instead of model weight, which can be seen from the expression of penalty IRM and penalty DAT:
>
> $\mathrm{Penalty_{IRM}} = ||\nabla_{w|w=1.0}R^e(w\cdot(\beta\cdot \Phi))||^2$ (Eq.4)
>
> $\mathrm{Penalty_{DAT}} = \left\|\nabla_x \mathbb E_{(x,y)\sim D_e}\ell(\beta^\top\Phi(x),y)\right\|$ (Eq.10)
>
> **First**, we can easily see that the two have a close connection, making DAT also potentially effective for addressing correlation-shift problems. **Second**, the gradient penalty of DAT is imposed in the input space while that of IRM is imposed on the weight space. In comparison, the input-space penalty could induce a better input landscape smoothing effect, which is shown to yield better domain transferability in recent works [1-2]. **Third**, this input-space regularization also enables DAT to remove domain-wise background information as analyzed in **Section 4**.
>
> [1] Do Adversarially Robust ImageNet Models Transfer Better?, https://arxiv.org/abs/2007.08489
>
> [2] Adversarially-Trained Deep Nets Transfer Better: Illustration on Image Classification, https://arxiv.org/abs/2007.05869
>
> ---
>
> Thanks for your constructive comments. Please let us know if you have additional questions.

---

> > ### Comment · Reviewer_Ywim · 2021-11-23
> > **Response to Authors**
> >
> > Thank you for your thorough response and the updated results! They are much appreciated.
> > Thanks also for the discussion and the results on CelebA and OfficeHome. It might be good to include those in the paper.
> >
> > I remain positive about the paper after seeing the author's response to mine and the other reviews.

---

> ### Author Response · Authors · 2021-11-20
> **Response to Reviewer Ywim (1/2)**
>
> Thank you for your positive comments on the novelty of this paper! Here is our answer to your concern.
>
> ---
>
> **Q1**: Results in Table 1 do not support the claim that AT may help performance under diversity-shift.
>
> **A1**: We note that due to the limit of time, we did not tune the hyperparameters of AT exhaustively at the time of submission. In our revision, we have updated the results of AT to **Table 1** as follows.
>
> | Algorithm | CMNIST           | NICO             | PACS             | TerraInc         | Avg      |
> | --------- | ---------------- | ---------------- | ---------------- | ---------------- | -------- |
> | ERM       | 58.5 +/- 0.3     | 71.4 +/- 1.3     | 81.5 +/- 0.0     | 42.6 +/- 0.9     | 63.5     |
> | AT        | 57.9 +/- 0.4     | 70.5 +/- 0.7     | **82.0 +/- 0.2** | 42.6 +/- 0.3     | 62.8     |
> | DAT       | **68.4 +/- 2.0** | **72.6 +/- 1.7** | **82.0 +/- 0.1** | **42.7 +/- 0.7** | **66.4** |
>
> From the table above, we can see that AT achieves **82.0%** on PACS, which indeed outperforms ERM, while only falling behind RSC and DAT. Instead, AT performs worse than the correlation-shift tasks, CMNIST and NICO. Therefore, this result justifies our analysis that AT is superior to ERM under diversity-shift and it is less effective under correlation shift. Apart from that, several previous studies [1,2] used the AT-like approach to account for diversity-shift, including PACS and corruption-type datasets, which also support our claim.
>
> [1] DecAug: Out-of-Distribution Generalization via Decomposed Feature Representation and Semantic Augmentation, https://arxiv.org/abs/2012.09382
>
> [2] Improved OOD Generalization via Adversarial Training and Pre-training, https://arxiv.org/abs/2105.11144
>
> ---
>
> **Q2**: The experiments of UAT are not presented.
>
> **A2**: We have added the following results of UAT algorithm to **Table 1** for comparison.
>
> | Algorithm | CMNIST           | NICO             | PACS             | TerraInc         | Avg      |
> | --------- | ---------------- | ---------------- | ---------------- | ---------------- | -------- |
> | ERM       | 58.5 +/- 0.3     | 71.4 +/- 1.3     | 81.5 +/- 0.0     | 42.6 +/- 0.9     | 63.5     |
> | UAT       | 58.7 +/- 2.3     | 69.1 +/- 1.2     | 80.7 +/- 0.4     | 41.8 +/- 1.2     | 62.6     |
> | DAT       | **68.4 +/- 2.0** | **72.6 +/- 1.7** | **82.0 +/- 0.1** | **42.7 +/- 0.7** | **66.4** |
>
> We can see that DAT outperforms UAT on all the datasets, and ERM has better performance than UAT on three of the four datasets, which highlights the necessity of using domain-wise perturbations to remove domain-varying background information, as discussed in **Section 4**.
>
> We have updated the UAT results in **Table 1** and included the experimental details in **Appendix A.2**.
>
> ---
>
> **Q3**: The datasets used are not as comprehensive as in Ye et al. (2021) or (Gulrajani & Lopez-Paz, 2021).
>
> **A3**: As stated in the paper, our algorithm has the property of suppressing the influence of background on classification results, which makes it more suitable for datasets with background information. Instead, OfficeHome and CelebA used in [1] do not fit this criterion well. For example, in OfficeHome, samples in both the Clipart and Product domains have no background at all; while for CelebA, it is **an artificially created dataset** where the domains are partitioned using face-related attributes, so there is no natural change in the background across domains. As a result, CelebA is not suitable for our DAT method, which is designed to remove domain-varying background information. Indeed, our DAT achieves inferior performance on these two datasets, with an accuracy of 85.9 (DAT) vs **87.2** (ERM) on CelebA and 61.1 (DAT) vs **63.3** (ERM) on OfficeHome. This shows that our DAT indeed relies on the prior that natural domain variations mostly happen in the background.
>
> [1] OoD-Bench: Benchmarking and Understanding Out-of-Distribution Generalization Datasets and Algorithms, https://arxiv.org/abs/2106.03721

---

### Author Response · Authors · 2021-11-20
**A Summary of Paper Updates**

We would like to express sincere gratitude towards all the anonymous reviews for their detailed and enlightening response and have made the following updates to address the reviewers’ comments.

1. **Section 2**: added previous works on AT and their connections to OOD robustness to related works.

2. **Section 2**: added more description and a figure to illustrate correlation shift and diversity shift.

3. **Section 4.2**: added the quantitative results of the performance of DAT and ERM on Mixed dataset.

4. **Section 5.1**: added more details on datasets and baselines.

5. **Section 5.1**: added more details on the pipeline of DAT.

6. **Section 5.2**: added UAT, WRM [1], and ADA [2]’s results to Table 1.

7. **Section 5.2**: updated the result of AT on PACS in Table 1.

8. **Appendix A.2**: added experimental details on WRM and ADA.

9. **Appendix A.2**: added domain-split results of ERM and UAT.



[1] Certifying Some Distributional Robustness with Principled Adversarial Training, https://arxiv.org/abs/1710.10571

[2] Generalizing to Unseen Domains via Adversarial Data Augmentation, https://arxiv.org/abs/1805.12018

---

> ### Author Response · Authors · 2021-11-29
> **General Response to All Reviewers**
>
> We want to sincerely thank all reviewers for their active responses in the discussion stage and for appreciating our feedbacks. We will append the corresponding discussions in the revision following the suggestions.

---

### Decision · Program_Chairs · 2022-01-20

**Decision:**

Reject

**Comment:**

This paper links OOD generalization with adversarial training and argues that adversarial training can help address the problem of OOD generalization. Based on all responses and reviews, there still are novelty concerns in this paper. In the meantime, this paper lacks theoretical justifications. More importantly, DAT only considers very limited situations regarding DG, which also reflects on its experimental results. In the following, I summarize the drawbacks of this paper for the possible revision in the future.

1. It seems not novel to link AT with OOD generalization since two reviewers show some references related to using AT to address DG.

2. From Eq. (9), DAT is based on perturbations rather than transformations. This means that DAT only considers very limited situations regarding DG. In the ordinary DG, source and target domains are from the same meta-distribution, which is clearly a more general case compared to the case considered in this paper. DAT-based AT might mislead the research direction of DG. It would be better to consider smart ways to generate adversarial examples, such as "Pixeldefend: Leveraging generative models to understand and defend against adversarial examples" (ICLR2018).

3. Eqs. (7), (8) and (10) are not rigours. It is not convincing to propose a method based on these formulas.

4. The method doesn't show substantial improvements compared to ERM in most tasks (CMNIST dominates the average), which implies the limitations of DAT (see 3).

5. There are no theoretical contributions regarding DG. This paper does not mention the key assumption behind the DAT. For example, DG can be a well-defined problem if source and target domains are from the same meta-distribution. However, this paper does not clarify what assumptions it assumes and does not show how DAT can address DG in theory.

Based on the above drawbacks, I recommend rejection for this paper.